# Does FLUX Already Know How to Perform Physically Plausible Image Composition?

**Shilin Lu**[1,*]**, Zhuming Lian**[1,*]**, Zihan Zhou**[1]**, Shaocong Zhang**[1]**,**
**Chen Zhao**[2]**, Adams Wai-Kin Kong**[1]
[1]Nanyang Technological University, [2]Nanjing University
{shilin002, zhuming001, zihan010, shaocong001}@e.ntu.edu.sg
602024710020@smail.nju.edu.cn, adamskong@ntu.edu.sg

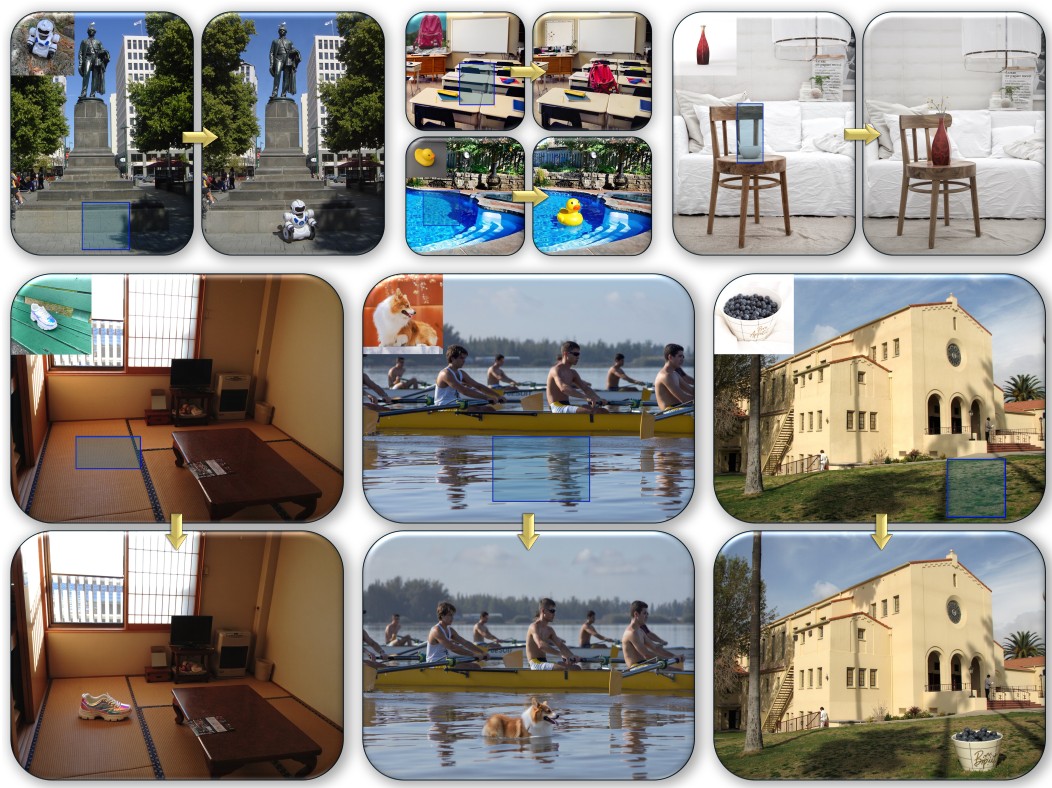

Figure 1: Showcase of our training-free image composition method, **SHINE**. This gallery highlights SHINE's ability to seamlessly integrate subjects into complex scenes, including **low-light conditions, intricate shadows, and water reflections.**

## ABSTRACT

Image composition aims to seamlessly insert a user-specified object into a new scene, but existing models struggle with complex lighting (e.g., accurate shadows, water reflections) and diverse, high-resolution inputs. Modern text-to-image diffusion models (e.g., SD3.5, FLUX) already encode essential physical and resolution priors, yet lack a framework to unleash them without resorting to latent inversion, which often locks object poses into contextually inappropriate orientations, or brittle attention surgery. We propose **SHINE**, a training-free framework for **S**eamless, **H**igh-fidelity **I**nsertion with **N**eutralized **E**rrors. SHINE introduces manifold-steered anchor loss, leveraging pretrained customization adapters (e.g., IP-Adapter) to guide latents for faithful subject representation while preserving background integrity. Degradation-suppression guidance and adaptive background blending are proposed to further eliminate low-quality outputs and visible seams. To address the lack of rigorous benchmarks, we introduce *ComplexCompo*, fea-

---

*Equal Contribution.

turing diverse resolutions and challenging conditions such as low lighting, strong illumination, intricate shadows, and reflective surfaces. Experiments on Complex-Compo and DreamEditBench show state-of-the-art performance on standard metrics (e.g., DINOv2) and human-aligned scores (e.g., DreamSim, ImageReward, VisionReward). Code is available at https://github.com/ZhumingLian/SHINE.

# 1 INTRODUCTION

Image composition, which places a user–specified object into a new scene, is a demanding image editing task. Despite the breathtaking progress of multimodal foundation models (e.g., GPT-5 (OpenAI, 2025), Gemini-2.5 (Gemini2.5, 2025), SeedEdit/Doubao (Shi et al., 2024b), and Grok-4 (gro, 2025)), these generic models still struggle with image composition. Typical failures include imprecise object placement, inconsistent lighting, and the subject's identity drift (see Fig. 2). These limitations indicate that, as of now, massive multimodal pre-training alone has not yet endowed them with sufficient compositional ability for this task. A natural response has been to train specialized models. Yet building large-scale, high-quality, multi-resolution triplet datasets (object, scene, composite) is prohibitively costly. As a result, most composition models are fine-tuned from base models (e.g., FLUX.1-dev (Black Forest Labs, 2024a), FLUX.1-Fill (Black Forest Labs, 2024c), SDXL (Podell et al., 2024)) on synthetic data generated via inpainting or augmentations (Chen et al., 2024c; Yang et al., 2023; Song et al., 2023; Wang et al., 2025a; He et al., 2024).

These models, however, face two main limitations (see Fig. 6): **(i) Lighting realism.** They struggle to achieve natural composition under complex lighting conditions, such as accurate shadow generation or water reflections for the inserted subject. **(ii) Resolution rigidity.** They are tied to a fixed resolution, necessitating downsampling or cropping when applied to varied, high-resolution background images, which degrades generation quality. Notably, such issues are absent in the base models, implying that the underlying physical priors are present but are not effectively exploited by fine-tuned variants. The degradation largely stems from low-quality synthetic datasets, which inherit flaws from inpainting models that often mis-handle shadows and reflections, producing implausible edits, hallucinated content, or incomplete object removal (Yu et al., 2025c; Winter et al., 2025).

There have been prior **training-free** attempts to exploit the priors of text-to-image (T2I) models for advancing image composition, but they fall short for two main reasons. **(i) Inversion bottlenecks.** Most methods (Lu et al., 2023d; Pham et al., 2024; Yan et al., 2025; Li et al., 2024b) depend on accurate image inversion (Song et al., 2021; Lu et al., 2022; Mokady et al., 2023). In practice, inversion constrains the inserted object to the pose of its reference image, often resulting in contextually inappropriate orientations. Moreover, inversion is less effective for classifier-free guidance (CFG) distilled models (e.g., FLUX), where elevated inversion errors degrade identity preservation. **(ii) Fragile attention surgery.** Many training-free approaches rely on attention manipulation (Lu et al., 2023d; Yan et al., 2025; Li et al., 2024b). While compatible with the joint self-attention in Multimodal Diffusion Transformers (MMDiT) (Peebles & Xie, 2023), these methods inherit the instability and hyperparameter sensitivity (Lu et al., 2023d), limiting their robustness.

To bridge these gaps we present **SHINE**, a training-free framework for **S**eamless, **H**igh-fidelity **I**nsertion with **N**eutralized **E**rrors (see Fig. 1). SHINE comprises three innovations: **(i) Manifold-Steered Anchor (MSA) loss**, which leverages pretrained open-domain customization adapters (e.g., IP-Adapter (Ye et al., 2023)) to steer noisy latents toward faithfully representing the reference subject while preserving the structural integrity of the background. **(ii) Degradation-Suppression Guidance (DSG)** that steers sampling away from low-quality distributions. **(iii) Adaptive Background Blending (ABB)** that eliminates visible seams along mask boundaries.

Existing benchmarks primarily comprise background images with a fixed resolution of $512 \times 512$ pixels. To evaluate performance across diverse, high-resolution, and demanding scenarios, we introduce *ComplexCompo*, a benchmark that includes varied resolutions, both landscape and portrait orientations, and complex conditions such as low lighting, intense illumination, intricate shadows, and water reflections. Extensive experiments on ComplexCompo and DreamEditBench (Li et al., 2023b) demonstrate that SHINE achieves state-of-the-art (SOTA) performance, surpassing baselines on standard metrics (e.g., DINOv2 (Oquab et al., 2024)) and human-aligned metrics (e.g., DreamSim (Fu et al., 2023), ImageReward (Xu et al., 2023), VisionReward (Xu et al., 2024)).

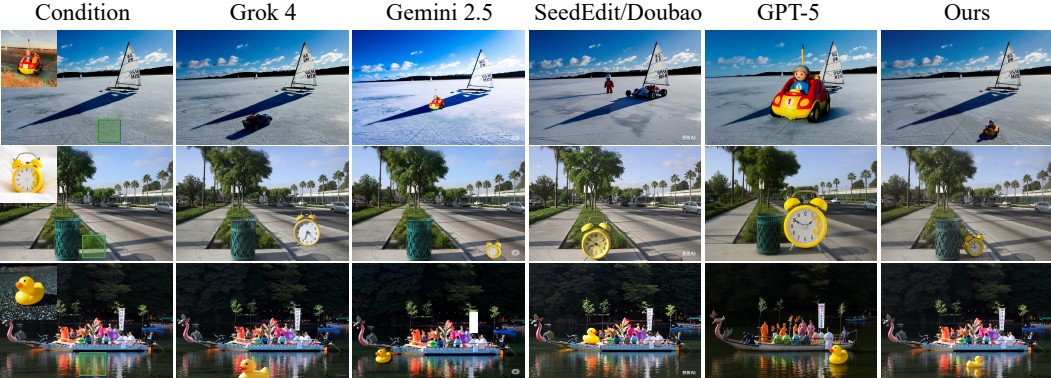

Figure 2: Image composition from advanced multimodal models under three challenging conditions: backlighting, shadows, and water surfaces. Refer to Appendix H for prompt details.

## 2 RELATED WORK

This section reviews prior work on image composition. A more comprehensive discussion, covering image composition, general image editing, and subject-driven generation, is offered in Appendix A. Classical image composition splits into sub-tasks (Niu et al., 2021) such as object placement (Azadi et al., 2020; Zhang et al., 2020a), blending (Wu et al., 2019; Zhang et al., 2020b), harmonization (Cao et al., 2023; Lu et al., 2023b), and shadow generation (Hong et al., 2022; Sheng et al., 2021), typically handled by separate models. Diffusion models have shifted the field toward unified frameworks, either training-based or training-free. Training-based approaches fine-tune diffusion models with curated datasets, adding grounding layers, controllability signals, or identity-preserving supervision from image or video sets (Wang et al., 2025a; Chen et al., 2024c; Yang et al., 2023; Song et al., 2023; Lu et al., 2023c). However, they often bias model priors and struggle with complex lighting due to the lack of large-scale real-world triplets. Training-free approaches avoid retraining by manipulating inversion and attention during inference, enabling flexible test-time adaptation (Yan et al., 2025; Li et al., 2024b; 2023b; Lu et al., 2023d; Pham et al., 2024). Yet these methods remain fragile: strong injections preserve identity but fix unnatural poses, while weaker ones improve realism at the cost of fidelity, reflecting a core trade-off between identity preservation and natural composition.

## 3 METHOD

Image composition seeks to integrate a subject into a designated area of a background image while preserving the integrity of the surrounding scene. This process typically requires three inputs: (1) one or more reference images of the subject $\{x_1^{\text{subj}}, x_2^{\text{subj}}, \ldots, x_n^{\text{subj}}\}$, (2) a background image $x^{\text{bg}}$, and (3) a user-provided mask $M^{\text{user}}$ specifying the insertion region within the background.

Our framework is built on three core components: Manifold-Steered Anchor (MSA) loss, Degradation-Suppression Guidance (DSG), and Adaptive Background Blending (ABB). Importantly, the design is model-agnostic and requires only standard features of modern generative models: MSA loss assumes that the base model supports either personalization finetuning or provides access to a pretrained personalization adapter, DSG uses self-attention maps, and ABB relies on text-image cross-attention. These mild assumptions enable seamless integration into existing pipelines without architectural changes. We present main results with FLUX, while additional experiments on SDXL (Podell et al., 2024), SD3.5 (Esser et al., 2024), and PixArt (Chen et al.) are provided in Appendix E. The complete algorithm is shown in Algorithm 1.

### 3.1 NON-INVERSION LATENT PREPARATION

In training-free diffusion-based image composition (Lu et al., 2023d; Pham et al., 2024; Yan et al., 2025; Li et al., 2024b), it is common to start from a noisy latent. Existing training-free frameworks typically rely on image inversion, where the initial noisy latent is constructed by copying the inverted latent of the subject image into a designated region of the background image's inverted latent.

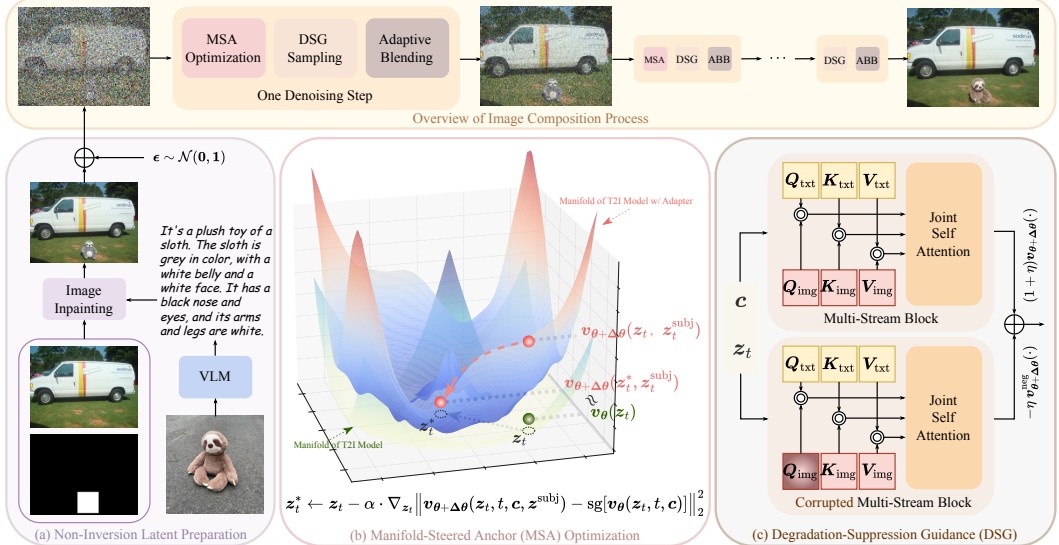

Figure 3: **Overview of the proposed framework. (a)** The noisy latent is created by inpainting the background with a VLM-derived object description, then adding Gaussian noise. **(b)** Manifold-Steered Anchor (MSA) loss guides noisy latents toward faithfully capturing the reference subject (red arrow), while preserving the structural integrity of the background. Concretely, it enforces that the prediction of the optimized latent $z_t^*$ on the adapter-augmented model's manifold remains close to the prediction of the original latent $z_t$ on the base model's manifold. **(c)** Degradation-Suppression Guidance (DSG) constructs a negative velocity pointing toward low-quality regions by blurring $Q_{\text{img}}$ and, in a CFG-like manner, steers the trajectory away from this low-quality distribution.

However, this copy-paste strategy constrains the inserted object to the exact pose of its reference image, often leading to contextually inappropriate orientations in the composed result. Moreover, inversion is suboptimal for CFG-distilled models (e.g., FLUX), as it introduces higher inversion errors that compromise subject identity preservation.

To address these limitations, we abandon inversion and instead perform a one-step forward diffusion to obtain the noisy latent. As illustrated in Fig. 3(a), we use a vision-language model (VLM) (Xue et al., 2024; Chen et al., 2024d; Liu et al., 2024) to caption the subject image and leverage this caption, along with an image inpainting model (Li et al., 2024c; Ju et al., 2024; Zhuang et al., 2024; Black Forest Labs, 2024b), to generate the image to which the subject is attached, denoted as $x^{\text{init}}$. The noisy latent is encoded in the VAE space as $z^{\text{init}}$ and perturbed to timestep $t \leq T$ via one-step forward diffusion, following the flow matching formulation: $z_t = (1 - \sigma_t)z^{\text{init}} + \sigma_t \epsilon$, where $\epsilon \sim \mathcal{N}(\mathbf{0}, \mathbf{1})$.

### 3.2 MANIFOLD-STEERED ANCHOR LOSS

The Manifold-Steered Anchor (MSA) loss is designed to optimize the noisy latent $z_t$ (from Sec. 3.1) during the denoising process, steering it toward a reference subject while preserving the structural integrity of the original image. The key intuition is to leverage the prior knowledge embedded in pretrained open-domain customization adapters (or alternatively, personalized LoRAs) such as IP-Adapter (Ye et al., 2023), PuLID (Guo et al., 2024), and InstantCharacter (Tao et al., 2025), to intervene directly in the diffusion trajectory. Specifically, the MSA loss is defined as:

$$\min_{z_t} \mathcal{L}_{\text{MSA}}(z_t) = \left\| v_{\theta + \Delta\theta}(z_t, t, c, z^{\text{subj}}) - \text{sg}[\tilde{v}_t] \right\|_2^2, \tag{1}$$

where $\tilde{v}_t \triangleq v_{\theta}(\tilde{z}_t, t, c)$ serves as a fixed anchor, preserving the structure of the background image at a given noise level $t$, with $\tilde{z}_t$ held constant as the original noisy latent. $v_{\theta}(\cdot)$ denotes the velocity predicted by the frozen T2I model $\theta$, while $v_{\theta + \Delta\theta}(\cdot)$ represents the velocity predicted by the a T2I model augmented with an adapter $\Delta\theta$. $z^{\text{subj}}$ is the latent of the subject image. The text prompt $c$ is from the VLM's description of $x^{\text{init}}$, and $\text{sg}[\cdot]$ indicates the stop-gradient operation.

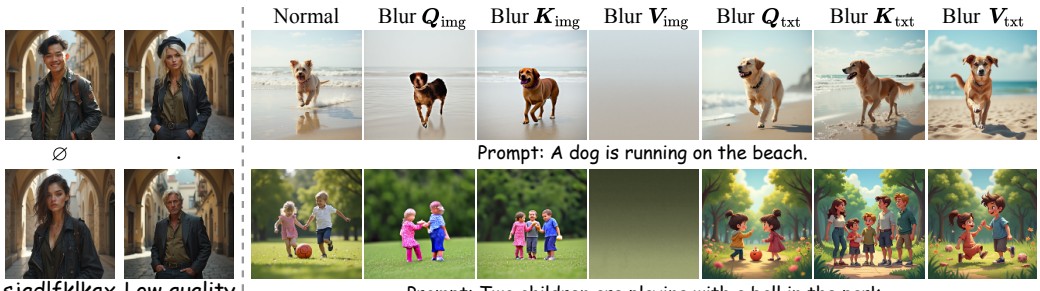

Figure 4: *Left:* Robustness of FLUX. *Right:* Impacts of blurring different features in FLUX.

The MSA loss is motivated by the observation that optimizing a latent representation against a frozen generative model implicitly projects the latent onto the model's learned data manifold (Meng et al., 2021; Kim et al., 2022; Graikos et al., 2022; Feng et al., 2023). The generator serves as an implicit prior, guiding gradient descent toward the manifold's basin of attraction.

For instance, when a generative model $G(w)$ is trained solely on cat images, its outputs are confined to the cat-image manifold. Thus, approximating a dog image $x_{\text{dog}}$ by solving $\min_w \|G(w) - x_{\text{dog}}\|_2^2$ yields $G(w^*)$ that remains a cat image, but with structural features (e.g., pose or outline) aligned to $x_{\text{dog}}$. The result is the projection of the dog image onto the cat manifold, not a genuine dog image.

Analogously, MSA loss is designed to achieve two goals simultaneously. (1) It seeks an optimized noisy latent $z_t^*$ that remains within the manifold of the adapter-augmented model when conditioned on the subject $z^{\text{subj}}$. (2) It encourages the adapter's prediction on this latent $z_t^*$ to align with the base model's prediction on the original latent $z_t$, i.e., $v_{\theta+\Delta\theta}(z_t^*, t, c, z^{\text{subj}}) \approx v_\theta(z_t, t, c)$ (see Fig. 3(b)). Since the velocity prediction of a T2I model on a noisy latent $z_t$ can also be interpreted as a coarse estimate of the clean image that encodes essential structural information (Zheng et al., 2023), this alignment preserves the spatial layout and background details inherited from the original image.

The gradient of $\mathcal{L}_{\text{MSA}}$ with respect to $z_t$ is:

$$\nabla_{z_t}\mathcal{L}_{\text{MSA}}(z_t) = 2\Big(v_{\theta+\Delta\theta}(z_t, t, c, z^{\text{subj}}) - \text{sg}[\tilde{v}_t]\Big)\frac{\partial v_{\theta+\Delta\theta}(z_t, t, c)}{\partial z_t}. \qquad (2)$$

The Jacobian term necessitates backpropagation through the MMDiT, which is computationally expensive. However, this scenario is analogous to Score Distillation Sampling (SDS) (Poole et al.), where research shows that omitting the Jacobian term yields an effective gradient for optimization with diffusion models. Thus, we adopt the same strategy for optimization.

### 3.3 DEGRADATION-SUPPRESSION GUIDANCE

MSA loss effectively facilitates the insertion of reference objects. However, due to the inherent stochasticity of the denoising and optimization process, the results sometimes suffer from degraded visual quality, manifesting as oversaturated colors and reduced identity consistency (see Fig. 5). To address this, we introduce Degradation-Suppression Guidance (DSG), inspired by negative prompting (Schramowski et al., 2023), defined as:

$$v_t^{\text{dsg}} = v_{\theta+\Delta\theta}(z_t, t, c, z^{\text{subj}}) + \eta\big(v_{\theta+\Delta\theta}(z_t, t, c, z^{\text{subj}}) - v_{\theta+\Delta\theta}^{\text{neg}}(z_t, t, c, z^{\text{subj}})\big), \qquad (3)$$

where $v_{\theta+\Delta\theta}^{\text{neg}}$ denotes a negative velocity prediction that guides the generation toward low-quality regions. A key challenge is the design of a meaningful negative velocity prediction $v_{\theta+\Delta\theta}^{\text{neg}}$ within MMDiT-based architectures. In our experiments with FLUX, we observed that using nonsensical text prompts or explicit negative prompts fails to introduce degradation. The generated images remain high-fidelity (Fig. 4(a)), suggesting that text-based negative prompting is ineffective for FLUX.

In our setting, the ideal negative velocity $v_{\theta+\Delta\theta}^{\text{neg}}$ should target directions that preserve the semantic content and spatial layout while lowering perceptual quality. To achieve this, we investigate whether we can manipulate FLUX's internal representations to construct such a targeted degradation signal.

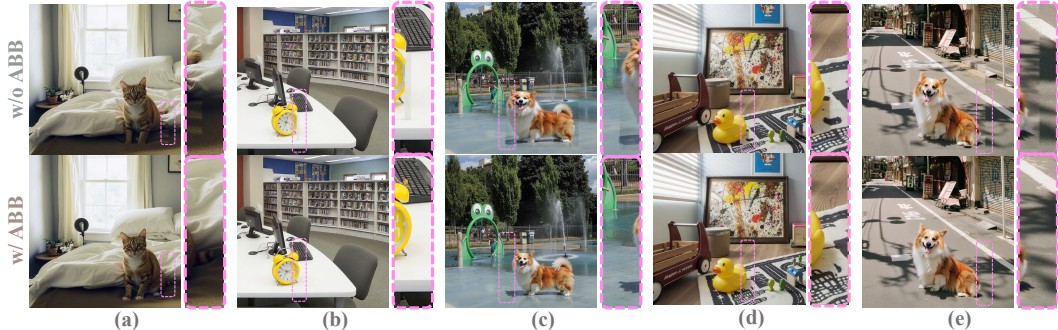

Figure 5: Comparison of rectangular-mask blending and Adaptive Background Blending (ABB). Boundary regions (pink dashed boxes) are enlarged for clarity. Zoom in for details.

In FLUX, both multi-stream and single-stream blocks compute joint self-attention over concatenated text and image tokens as follows:

$$h = \text{softmax}\Big([\boldsymbol{Q}_{\text{txt}}, \boldsymbol{Q}_{\text{img}}][\boldsymbol{K}_{\text{txt}}, \boldsymbol{K}_{\text{img}}]^\top / \sqrt{d_k}\Big) \cdot [\boldsymbol{V}_{\text{txt}}, \boldsymbol{V}_{\text{img}}], \tag{4}$$

where $[\boldsymbol{Q}_{\text{txt}}, \boldsymbol{Q}_{\text{img}}]$ represents the concatenation of text and image queries, and similarly for keys and values. To identify an effective manipulation strategy, we systematically perturb different components in the attention mechanism (i.e., $\boldsymbol{Q}_{\text{txt}}, \boldsymbol{K}_{\text{txt}}, \boldsymbol{V}_{\text{txt}}, \boldsymbol{Q}_{\text{img}}, \boldsymbol{K}_{\text{img}}$ and $\boldsymbol{V}_{\text{img}}$) and observe their impact on generation quality.

As shown in Fig. 4(b), our findings are as follows:

1. Blurring $\boldsymbol{Q}_{\text{txt}}, \boldsymbol{K}_{\text{txt}}$, or $\boldsymbol{V}_{\text{txt}}$ has negligible impact on semantic fidelity and visual quality.
2. Blurring $\boldsymbol{V}_{\text{img}}$ severely disrupts the output distribution, leading to unintelligible images.
3. Blurring $\boldsymbol{K}_{\text{img}}$ moderately impacts quality, while the image remains visually acceptable.
4. Blurring $\boldsymbol{Q}_{\text{img}}$ yields pronounced degradations while preserving structural integrity, making it the most effective lever for constructing a negative velocity.

Based on these insights, we construct the negative velocity prediction $\boldsymbol{v}^{\text{neg}}_{\theta+\Delta\theta}$ in Eqn. 3 by blurring $\boldsymbol{Q}_{\text{img}}$ within FLUX (see Fig. 3(c)). Moreover, we show that blurring $\boldsymbol{Q}_{\text{img}}$ is mathematically equivalent to blurring the self-attention weights, whereas blurring $\boldsymbol{K}_{\text{img}}$ or $\boldsymbol{V}_{\text{img}}$ is not (see Appendix C for the proof). This equivalence is consistent with the fact that attenuating self-attention activations suppresses informative interactions and thus degrades image quality (Lu et al., 2024a).

### 3.4 ADAPTIVE BACKGROUND BLENDING

Previous methods typically rely on the user-provided mask $\boldsymbol{M}^{\text{user}}$ to preserve the background during each denoising step, blending as $\boldsymbol{z}'_t = \boldsymbol{M}^{\text{user}} \odot \boldsymbol{z}_t + (1 - \boldsymbol{M}^{\text{user}}) \odot \boldsymbol{z}^{\text{bg}}_t$, but this often introduces visible seams along mask boundaries (see the first row of Fig. 5).

To address this limitation, we propose Adaptive Background Blending (ABB), defined as

$$\boldsymbol{z}'_t = \hat{\boldsymbol{M}} \odot \boldsymbol{z}_t + \Big(1 - \hat{\boldsymbol{M}}\Big) \odot \boldsymbol{z}^{\text{bg}}_t, \qquad \hat{\boldsymbol{M}} = \mathbb{1}\{t > \tau\}\,\mathcal{D}(\boldsymbol{M}^{\text{attn}}) + \mathbb{1}\{t \le \tau\}\,\boldsymbol{M}^{\text{user}}, \tag{5}$$

where $\boldsymbol{M}^{\text{user}}$ is the user mask, while $\boldsymbol{M}^{\text{attn}}$ is derived by binarizing the cross-attention maps corresponding to subject tokens. These maps can be obtained by either averaging across layers or selecting the most informative layer via a lightweight analysis (details in Appendix D). The operator $\mathcal{D}(\cdot)$ performs dilation and extracts the largest connected component, ensuring robustness to noise.

Compared to $\boldsymbol{M}^{\text{user}}$, $\boldsymbol{M}^{\text{attn}}$ is more spatially precise, particularly for elongated or irregularly shaped objects that do not fully occupy a rectangular region. As illustrated in the second row of Fig. 5, our method produces smoother transitions by replacing the rigid user mask with the semantically guided mask. This refinement better preserves the surrounding scene, enabling seamless integration between generated content and the original background. However, applying this method throughout

---

**Algorithm 1** The Image Composition Process of SHINE.

---

**Input:** A background latent $z^{\text{bg}} = \text{VAE}(x^{\text{bg}})$, a subject latent $z^{\text{subj}} = \text{VAE}(x^{\text{subj}})$, a inpainted latent $z^{\text{init}} = \text{VAE}\left(\text{Inpainting}\left[x^{\text{bg}}, M^{\text{user}}, \text{VLM}(x^{\text{subj}})\right]\right)$, a user mask $M^{\text{user}}$.
**Output:** The composition latent $z_0$.

---

1: $z_{t_1} \leftarrow (1 - \sigma_{t_1})z^{\text{init}} + \sigma_{t_1}\epsilon$, where $\epsilon \sim \mathcal{N}(\mathbf{0}, \mathbf{1})$
2: **for** $t = t_1, \ldots, 0$ **do**
3:     `// Manifold-Steered Anchor (MSA) Optimization`
4:     **if** $t > \tau$ **then**
5:         $\tilde{v}_t \leftarrow v_\theta(z_t, t, c)$
6:         **for** $j = 1, \ldots, k$ **do**
7:             $z_t \leftarrow z_t - \alpha \cdot M^{\text{user}} \odot \nabla_{z_t} \left\| v_{\theta + \Delta\theta}(z_t, t, c, z^{\text{subj}}) - \text{sg}[\tilde{v}_t] \right\|_2^2$
8:         **end for**
9:     **end if**
10:     `// Degradation-Suppression Guidance (DSG)`
11:     $v_t, A_t \leftarrow v_{\theta + \Delta\theta}(z_t, t, c, z^{\text{subj}})$
12:     $v_t^{\text{dsg}} \leftarrow v_t + \eta \left(v_t - v_{\theta + \Delta\theta}^{\text{neg}}(z_t, t, c, z^{\text{subj}})\right)$
13:     $z_{t-1} \leftarrow z_t + (\sigma_{t-1} - \sigma_t)v_t^{\text{dsg}}$
14:     $z_{t-1}^{\text{bg}} \leftarrow (1 - \sigma_{t-1})z^{\text{bg}} + \sigma_{t-1}\epsilon$, where $\epsilon \sim \mathcal{N}(\mathbf{0}, \mathbf{1})$
15:     `// Adaptive Background Blending (ABB)`
16:     $M^{\text{attn}} \leftarrow \text{MaxConnectedComponent}\left(\text{Dilate}(\mathbb{1}(A_t \geq \gamma))\right)$
17:     $\hat{M} \leftarrow \mathbb{1}\{t > \tau\} M^{\text{attn}} + \mathbb{1}\{t \leq \tau\} M^{\text{user}}$
18:     $z_{t-1} \leftarrow \hat{M} \odot z_{t-1} + \left(1 - \hat{M}\right) \odot z_{t-1}^{\text{bg}}$
19: **end for**
20: **return** $z_0$

---

the denoising process may truncate object shadows or reflections. Through empirical evaluation, we find that leveraging $M^{\text{attn}}$ during the initial denoising steps $(t > \tau)$ sufficiently mitigates visible seams along mask boundaries, ensuring high-fidelity scene coherence.

## 4 EXPERIMENTS

### 4.1 EXPERIMENTAL SETUP

**Benchmark.** Current benchmarks primarily consist of background images with a fixed resolution of $512 \times 512$ pixels. To assess performance across diverse, high-resolution, and complex scenarios, we introduce *ComplexCompo*, a benchmark built upon DreamEditBench (Li et al., 2023b). DreamEdit-Bench includes 220 (subject, background, bounding box) pairs designed for $512 \times 512$ resolution. In contrast, ComplexCompo features 300 composition pairs with varying resolutions, encompassing both landscape and portrait orientations, and incorporates challenging conditions such as low lighting, intense illumination, intricate shadows, and water reflections. The background images are sourced from OpenImage (Kuznetsova et al., 2020). Further details are provided in Appendix G.

**Metrics.** Previous methods primarily adapt CLIP-I (Radford et al., 2021) and DINOv2 (Oquab et al., 2024) to assess subject identity consistency. However, these features capture high-level semantic information that may not fully align with human perception of finer-grained attributes. Thus, we further incorporate instance retrieval features (IRF) from (Shao & Cui, 2022) and DreamSim (Fu et al., 2023), which better align with human judgments. An analysis of identity consistency metrics is provided in Appendix I. For overall image quality, we use ImageReward (IR) (Xu et al., 2023) and VisionReward (VR) (Xu et al., 2024), fine-grained reward models that more accurately reflect human preferences. To more comprehensively evaluate composition quality, we further include three UnifiedReward variants (Wang et al., 2025c;d) and HPSv3 (Ma et al., 2025b) in Appendix L. Background consistency is measured using LPIPS (Zhang et al., 2018) and SSIM (Wang et al., 2004).

**Implementation Details.** In our experiment, we used FLUX.1-dev, a 12B-parameter flow matching model, as the base model, combined with InstantCharacter (Tao et al., 2025) as the adapter (Addi-

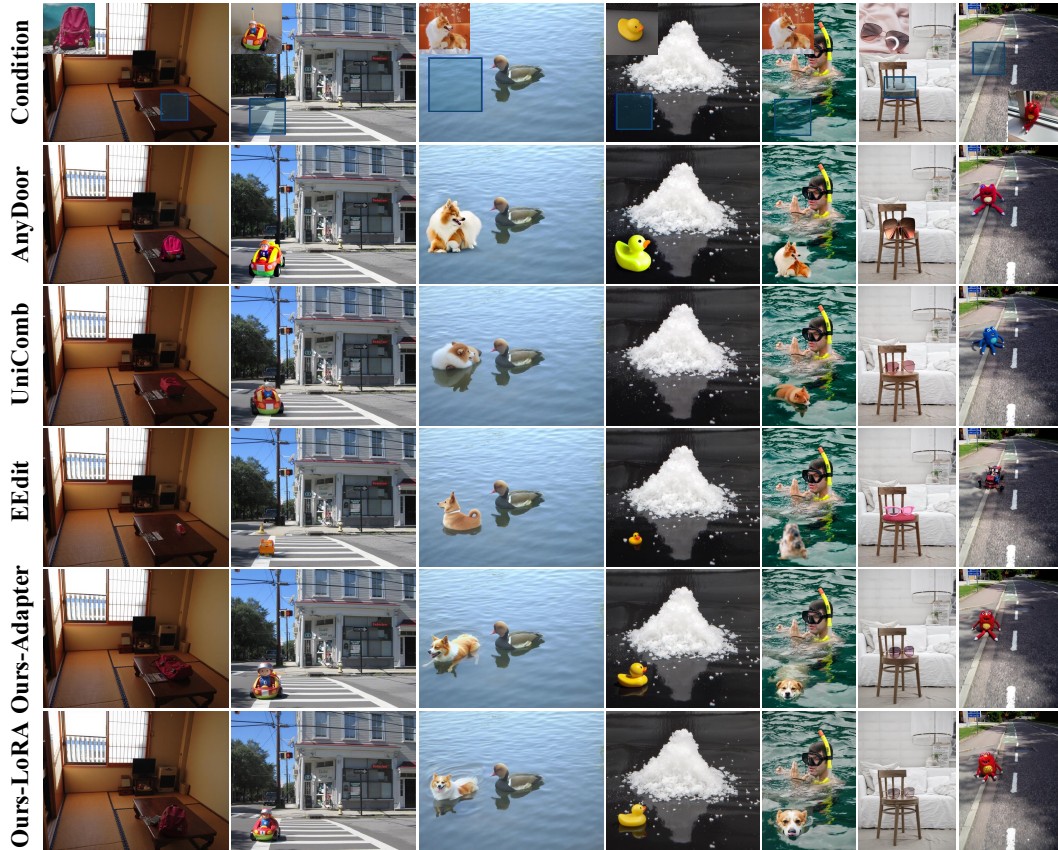

Figure 6: Qualitative comparison of our method with multiple baselines in challenging scenarios, **drawn from our benchmark dataset**. More qualitative comparisons are available in Appendix O.

tional results on SDXL, SD3.5, and PixArt are presented in Appendix E). Our approach also supports per-concept LoRA (Hu et al.), which requires test-time tuning (Ruiz et al., 2023) but delivers superior identity consistency compared to an open-domain adapter, making it ideal for scenarios demanding precise identity preservation. The denoising schedule consists of 20 steps, with the inpainted image perturbed to timestep 15 and denoising initiated from that point. We use Flux.1 Fill (Black Forest Labs, 2024b) as the inpainting model and BLIP-3 (Xue et al., 2024) as the VLM. Additional details and hyperparameters are provided in Appendix J.

## 4.2 EXPERIMENTAL RESULTS

We compare our method with two categories of baselines: **(1) Training-based methods (6 in total)**: UniCombine (Wang et al., 2025a), AnyDoor (Chen et al., 2024c), Paint by Example (PBE) (Yang et al., 2023), ObjectStitch (Song et al., 2023), MADD (He et al., 2024), and DreamCom (Lu et al., 2023c); **(2) Training-free methods (5 in total)**: EEdit (Yan et al., 2025), TIGIC (Li et al., 2024b), DreamEdit (Li et al., 2023b), TF-ICON (Lu et al., 2023d), and TALE (Pham et al., 2024).

As shown in Tab. 1, both variants of our method surpass all baselines on DreamEditBench across human preference aligned metrics (i.e., DreamSim, IR, VR), which are the most critical indicators of quality. For background-related metrics, all methods achieve comparable results, with differences so small they are imperceptible to the human eye. On the more challenging ComplexCompo dataset, which includes non-square resolutions and intricate scenes, most methods experience a notable performance drop, yet our approach consistently remains the top performer. From Fig. 6, it is evident that while AnyDoor achieves high scores on many identity metrics, the model tends to copy and paste the subject into the scene, resulting in unnatural compositions and lower image quality scores. In contrast, our method excels at naturally composing objects in challenging conditions (e.g., low-light settings, water surfaces, and scenes with complex shadows). Appendix F provides user study.

Table 1: Comparison of composition performance across two benchmarks. The best result in each column is highlighted in **bold**, while the second-best is underlined. Metrics shown in pink are those specifically trained to better align with human preferences. Abbreviations: IRF= Instance Retrieval Features; IR = ImageReward; VR = VisionReward; URE = UnifiedReward-Edit-qwen3vl-8b.

| Bench | Method | Training-Free | Base Model | External Model | Subject Identity Consistency | | | | Background | | Image Quality | | | |
|---|---|---|---|---|---|---|---|---|---|---|---|---|---|---|
| | | | | | CLIP-I↑ | DINOv2↑ | IRF↑ | DreamSim↓ | LPIPS↓ | SSIM↑ | IR↑ | VR↑ | HPS↑ | URE↑ |
| Dream-Edit-Bench (220) | Flux.1 Fill (Black Forest Labs, 2024b) | ✗ | FLUX | - | 0.7328 | 0.6745 | 0.5754 | 0.5233 | 0.0166 | 0.9076 | 0.5577 | 3.5997 | 8.6432 | 21.5812 |
| | MADD (He et al., 2024) | ✗ | SD | DINO | 0.7118 | 0.6279 | 0.4333 | 0.5810 | 0.0604 | 0.8182 | -0.2545 | 2.7011 | 1.2443 | 13.8148 |
| | ObjectStitch (Song et al., 2023) | ✗ | SD | VIT | 0.7567 | 0.6930 | 0.5525 | 0.5093 | 0.0190 | 0.8316 | 0.0791 | 3.2416 | 7.4529 | 19.1886 |
| | DreamCom (Lu et al., 2023c) | ✗ | SD | LoRA | 0.7414 | 0.6749 | 0.5597 | 0.5626 | 0.0200 | 0.8283 | 0.1873 | 3.5053 | 5.9324 | 19.9296 |
| | AnyDoor (Chen et al., 2024c) | ✗ | SD | DINO | **0.8183** | 0.7283 | 0.7714 | 0.3764 | 0.0251 | 0.8894 | 0.4511 | 3.3946 | 8.4867 | 19.0989 |
| | UniCombine (Wang et al., 2025a) | ✗ | FLUX | LoRA | 0.8058 | 0.7332 | 0.7579 | 0.3984 | 0.0050 | 0.9397 | 0.4565 | 3.6108 | 8.8415 | 21.7080 |
| | PBE (Yang et al., 2023) | ✗ | SD | - | 0.7742 | 0.7040 | 0.5845 | 0.4985 | 0.0197 | 0.8287 | 0.2083 | 3.3482 | 8.3789 | 20.2137 |
| | TIGIC (Li et al., 2024b) | ✔ | SD | - | 0.7226 | 0.6718 | 0.4711 | 0.6108 | 0.0584 | 0.8153 | -0.1332 | 2.9873 | 5.2676 | 17.1000 |
| | TALE (Pham et al., 2024) | ✔ | SD | - | 0.7329 | 0.6604 | 0.5007 | 0.6176 | 0.0392 | 0.8251 | -0.1502 | 3.1349 | 6.3773 | 18.0784 |
| | TF-ICON (Lu et al., 2023d) | ✔ | SD | - | 0.7479 | 0.6865 | 0.5179 | 0.5441 | 0.0582 | 0.8111 | 0.0816 | 3.2823 | 7.2643 | 18.2716 |
| | DreamEdit (Li et al., 2023b) | ✔ | SD | LoRA, VIT | 0.7703 | 0.7151 | 0.6147 | 0.5047 | 0.0140 | **0.9775** | 0.1744 | 3.1775 | 6.0250 | 15.7636 |
| | EEdit (Yan et al., 2025) | ✔ | FLUX | - | 0.6998 | 0.6590 | 0.4438 | 0.6160 | **0.0039** | 0.9475 | 0.0216 | 3.3606 | 6.6689 | 19.5603 |
| | Ours-Adapter | ✔ | FLUX | Adapter | 0.8086 | 0.7415 | 0.7702 | 0.3730 | 0.0236 | 0.8959 | 0.5709 | **3.6234** | **8.8861** | **22.0182** |
| | Ours-LoRA | ✔ | FLUX | LoRA | 0.8125 | 0.7452 | 0.7900 | 0.3577 | 0.0271 | 0.8847 | **0.5906** | 3.6161 | 8.8688 | 21.9421 |
| Complex-Compo (300) | Flux.1 Fill (Black Forest Labs, 2024b) | ✗ | FLUX | - | 0.7108 | 0.6475 | 0.5466 | 0.6018 | 0.0232 | 0.7442 | 0.4088 | 3.5737 | 8.7376 | 19.7712 |
| | MADD (He et al., 2024) | ✗ | SD | DINO | 0.6780 | 0.5993 | 0.3638 | 0.5979 | 0.0781 | 0.5658 | -0.0088 | 2.6582 | 5.9673 | 13.0567 |
| | ObjectStitch (Song et al., 2023) | ✗ | SD | VIT | 0.7608 | 0.7077 | 0.5513 | 0.4717 | 0.0388 | 0.6357 | 0.2482 | 3.4411 | 8.8389 | 18.8283 |
| | DreamCom (Lu et al., 2023c) | ✗ | SD | LoRA | 0.648 | 0.5692 | 0.2788 | 0.8192 | 0.0389 | 0.6342 | -0.0778 | 3.3409 | 7.9884 | 18.6143 |
| | AnyDoor (Chen et al., 2024c) | ✗ | SD | DINO | 0.7982 | 0.7052 | 0.7319 | 0.4493 | 0.0299 | 0.7262 | 0.3804 | 3.3787 | 8.9760 | 18.3550 |
| | UniCombine (Wang et al., 2025a) | ✗ | FLUX | LoRA | 0.7361 | 0.6552 | 0.5380 | 0.5682 | 0.0237 | 0.7077 | 0.2470 | 3.5454 | 8.8999 | 19.8529 |
| | PBE (Yang et al., 2023) | ✗ | SD | - | 0.7537 | 0.6802 | 0.5189 | 0.5187 | 0.0397 | 0.6321 | 0.2139 | 3.4310 | 8.5923 | 18.9507 |
| | TIGIC (Li et al., 2024b) | ✔ | SD | - | 0.6913 | 0.6329 | 0.3848 | 0.6549 | 0.0929 | 0.6228 | -0.131 | 2.8898 | 7.6630 | 16.4301 |
| | TALE (Pham et al., 2024) | ✔ | SD | - | 0.6816 | 0.6151 | 0.3799 | 0.6773 | 0.059 | 0.6334 | 0.0783 | 3.3498 | 8.7351 | 18.7567 |
| | TF-ICON (Lu et al., 2023d) | ✔ | SD | - | 0.6987 | 0.6435 | 0.4167 | 0.6030 | 0.0815 | 0.6216 | 0.1798 | 3.4323 | 9.3258 | 18.2366 |
| | DreamEdit (Li et al., 2023b) | ✔ | SD | LoRA, VIT | 0.7314 | 0.6722 | 0.5069 | 0.5670 | 0.0468 | 0.7201 | 0.1212 | 3.2531 | 8.0434 | 15.8934 |
| | EEdit (Yan et al., 2025) | ✔ | FLUX | - | 0.6713 | 0.6153 | 0.3797 | 0.6821 | **0.0226** | 0.7107 | 0.1433 | 3.5009 | 8.7835 | 19.7348 |
| | Ours-Adapter | ✔ | FLUX | Adapter | 0.7721 | 0.7107 | 0.6764 | 0.4294 | 0.0404 | 0.7789 | 0.4090 | **3.6020** | 9.6485 | 20.7349 |
| | Ours-LoRA | ✔ | FLUX | LoRA | **0.7999** | **0.7384** | **0.7659** | **0.3542** | 0.0430 | 0.7634 | **0.4246** | 3.5951 | **9.8418** | 21.0326 |

Table 2: Ablation study examining the impact of key components on DreamEditBench.

| Method | MSA | DSG | ABB | Subject Identity Consistency | | | | Background | | Image Quality | |
|---|---|---|---|---|---|---|---|---|---|---|---|
| | | | | CLIP-I | DINOv2 | IRF | DreamSim | LPIPS | SSIM | IR | VR |
| Config A | ✗ | ✗ | ✗ | 0.7328 | 0.6745 | 0.5754 | 0.5233 | **0.0166** | 0.9076 | 0.5577 | 3.5997 |
| Config B | ✔ | ✗ | ✗ | 0.7814 | 0.7204 | 0.7414 | 0.3951 | 0.0172 | 0.9075 | 0.5455 | 3.5952 |
| Config C | ✗ | ✔ | ✗ | 0.7528 | 0.6941 | 0.6533 | 0.4436 | 0.0178 | 0.9038 | 0.5633 | 3.6130 |
| Config D | ✗ | ✗ | ✔ | 0.7421 | 0.6814 | 0.6158 | 0.5127 | 0.0210 | 0.9010 | 0.5595 | 3.6109 |
| Config E | ✗ | ✔ | ✔ | 0.7481 | 0.6987 | 0.6647 | 0.4317 | 0.0218 | 0.8971 | **0.5850** | **3.6277** |
| Config F | ✔ | ✗ | ✔ | 0.8084 | **0.7429** | 0.7609 | 0.3756 | 0.0231 | 0.8991 | 0.5459 | 3.6023 |
| Config G | ✔ | ✔ | ✗ | 0.8077 | 0.7375 | 0.7589 | 0.3762 | 0.0182 | 0.9037 | 0.5745 | 3.6191 |
| Ours-Adapter | ✔ | ✔ | ✔ | **0.8086** | 0.7415 | **0.7702** | **0.3730** | 0.0236 | 0.8959 | 0.5709 | 3.6232 |

## 4.3 ABLATION STUDY

We validate our design choices through ablation (see Tab. 2 and Fig. 7). The results highlight three key insights. First, MSA loss notably improves subject identity consistency. Second, DSG boosts IR and VR scores by steering denoising away from low-quality regions. Finally, ABB effectively suppresses visible seams along mask boundaries. While this improvement is readily apparent in visual comparisons (Figs. 5, 7), it is less well captured by quantitative metrics, since LPIPS and SSIM primarily assess structural similarity rather than perceptual smoothness.

## 5 CONCLUSION

We introduced SHINE, a training-free framework for seamless and high-fidelity image composition with pretrained T2I models. SHINE integrates Manifold-Steered Anchor Loss, Degradation-Suppression Guidance, and Adaptive Background Blending to ensure precise subject placement and artifact-free synthesis across diverse resolutions and lighting conditions. To enable rigorous evaluation, we proposed ComplexCompo, a benchmark for challenging composition scenarios. SHINE achieves state-of-the-art results on both ComplexCompo and DreamEditBench.

**Limitations.** Our method reliably converges to the correct subject identity through MSA optimization even when the inpainted subject's appearance deviates substantially from the reference (see Fig. 8(a)). However, when the inpainting prompt specifies an incorrect color, the final inpainted result tends to inherit and preserve this erroneous color (see Fig. 8(b)). On the other hand, the similarity between the inserted object and the user-provided object depends on the quality of the customization adapter used. As shown in Tab. 1, because LoRA performs test-time tuning for individual concepts, it generates subjects that are more similar to the target than those produced by pretrained open-domain customization adapters, resulting in higher subject identity consistency metrics in the composition. While current customization adapters already perform well, the potential of our method will continue to improve as advancements are made in the field of open-domain customization adapters.

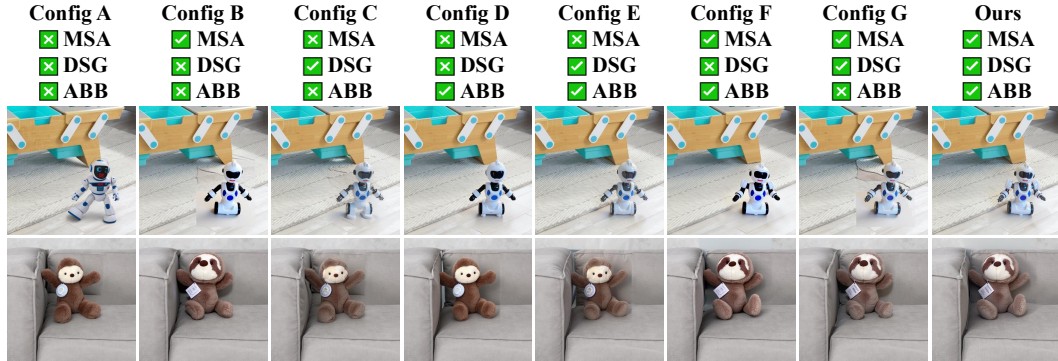

Figure 7: Qualitative ablation study comparing different variants of our framework.

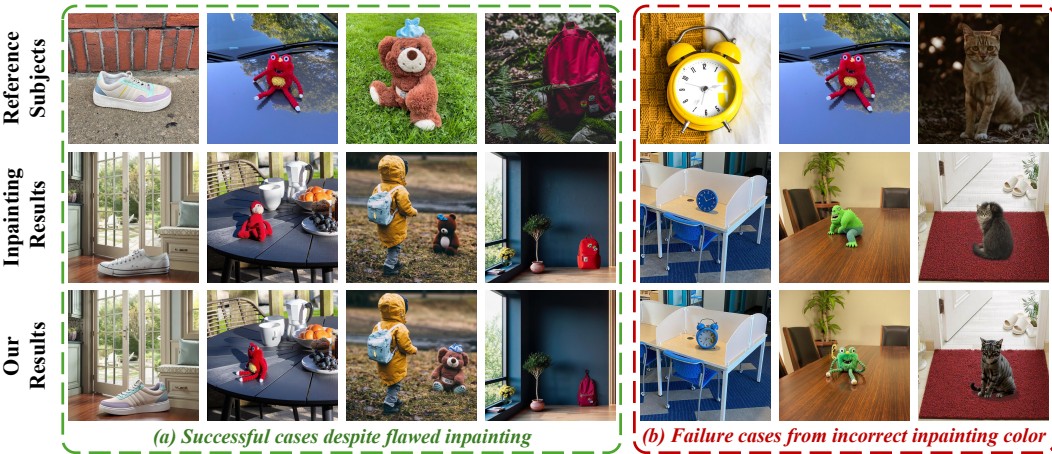

Figure 8: Composition inherit erroneous colors if the inpainting prompt specifies an incorrect color.

## ETHICS STATEMENT

Our framework provides an accessible way for people without professional artistic skills to create image compositions. While this technology offers significant benefits, it also carries the risk of misuse for malicious purposes, such as harassment or spreading misinformation. Additionally, our framework relies on pretrained large-scale T2I models, which may inadvertently introduce social and cultural biases. Therefore, using these models raises ethical concerns and requires careful consideration. We therefore urge users to exercise caution and use this tool responsibly for appropriate purposes only.

## REPRODUCIBILITY STATEMENT

We have taken several measures to ensure the reproducibility of our results. Algorithm 1 presents the pseudocode of our method. Details of the implementation for the main experiments are provided in Sec. 4.1, while the hyperparameter configurations are listed in Appendix J. The source code and the ComplexCompo dataset will be released publicly. This work adheres to the reproducibility requirements set by ICLR.

## ACKNOWLEDGMENTS

This research is supported by the National Research Foundation, Singapore and Infocomm Media Development Authority under its Trust Tech Funding Initiative. Any opinions, findings and conclusions or recommendations expressed in this material are those of the author(s) and do not reflect the views of National Research Foundation, Singapore and Infocomm Media Development Authority.

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

# Appendix

## Table of Contents

## A  RELATED WORK

### A.1  IMAGE COMPOSITION

Image composition involves integrating specific objects and scenarios from user-provided photos, often guided by text prompts. Traditionally, this process is divided into sub-tasks (Niu et al., 2021), including object placement (Azadi et al., 2020; Chen & Kae, 2019; Lin et al., 2018; Tripathi et al., 2019; Zhang et al., 2020a), image blending (Wu et al., 2019; Zhang et al., 2020b), image harmonization (Cao et al., 2023; Lu et al., 2023b; Zhang et al., 2020b; Cong et al., 2020; Cun & Pun, 2020; Jiang et al., 2021; Xue et al., 2022), and shadow generation (Hong et al., 2022; Liu et al., 2020; Sheng et al., 2021; Zhang et al., 2019), each typically handled by distinct models and pipelines. However, with the rise of diffusion-based generative models, recent approaches have shifted toward unified frameworks that address all sub-tasks simultaneously. These methods are broadly classified into training-based and training-free approaches.

**Training-based methods** fine-tune foundational models using datasets tailored for image composition. Early methods like Paint by Example (Yang et al., 2023) and ObjectStitch (Song et al., 2023) use CLIP to encode subject features, ensuring high semantic similarity between inserted objects and reference images. These approaches use image augmentation to create training datasets, enabling effective training. GLIGEN (Li et al., 2023c) incorporates grounding information into new trainable layers of a pre-trained diffusion model via a gated mechanism. ControlCom (Zhang et al., 2023a) integrates 2-dim indicator vector to improve controllability. DreamCom (Lu et al., 2023c) and Mure-ObjectStitch (Chen et al., 2024a) fine-tune models with small sets of reference images to preserve subject identity. AnyDoor (Chen et al., 2024c), IMPRINT (Song et al., 2024), and E-MD3C (Pham et al., 2025) leverage DINOv2 to enhance identity fidelity and control over shape and pose, drawing supervision from video data. MimicBrush (Chen et al., 2024b) similarly uses video-derived supervision for imitative editing. In contrast, MADD (He et al., 2024), ObjectMate (Winter et al., 2025), and OmniPaint (Yu et al., 2025c) employ image inpainting models to generate higher-quality training datasets compared to those based on image or video augmentation. Multitwine (Tarrés et al., 2025) enables the integration of multiple objects, capturing interactions from simple positional relationships to complex actions requiring reposing. DreamFuse (Huang et al., 2025) uses a Positional Affine mechanism to embed foreground size and position into the background, fostering effective foreground-background interaction through shared attention. Insert Anything (Song et al., 2025a) and UniCombine (Wang et al., 2025a) introduces a FLUX-based, multi-conditional generative framework that handles diverse condition combinations. However, these methods often bias the generative priors of base models toward curated datasets, resulting in unnatural compositions, such as implausible object-environment interactions (e.g., missing or unrealistic shadows and reflections). This stems from the absence of a large-scale, high-quality, multi-resolution, real-world triplet dataset comprising an object, a scene, and the object seamlessly integrated into the scene, which is expensive to produce.

**Training-free methods**, on the other hand, modify the inference process of pre-trained models to achieve composition without additional training. Early approaches like TF-ICON (Lu et al., 2023d) leverage accurate image inversion to lay the groundwork for composition, achieved through composite self-attention map injection. TALE (Pham et al., 2024) and PrimeComposer (Wang et al., 2024c) build on TF-ICON to enhance identity preservation and background-object style adaptation. TIGIC (Li et al., 2024b) focuses on preserving non-target areas during composition. Thinking Outside the BBox (Canet Tarrés et al., 2024) enables unconstrained image compositing, unbound by input masks. FreeCompose (Chen et al., 2024e) employs a pipeline of object removal, image harmonization, and semantic composition. DreamEdit (Li et al., 2023b), UniCanvas (Jin et al., 2025), and Magic Insert (Ruiz et al., 2025) use test-time tuning to fine-tune models during inference. Add-it (Tewel et al.) enables text-guided object insertion on FLUX, where users describe objects via text prompts instead of reference images. EEdit (Yan et al., 2025) recently improves TF-ICON on FLUX, introducing step-skipping to reduce time costs and spatial locality caching to minimize redundancy. However, training-free methods rely on precise image inversion and fragile attention surgery, which can lock inserted objects into the exact pose of the reference image, leading to awkward or contextually inappropriate orientations. Attention manipulation often causes instability and hyperparameter sensitivity, as feature or attention map injection does not always preserve subject identity. This creates a trade-off: stronger injection preserves identity but results in unnatural poses, while lighter injection yields more natural poses but compromises identity.

## A.2   GENERAL IMAGE EDITING

Recent and significant advancements in text-to-image generative models have enhanced numerous applications (Zhou et al., 2023; 2024a;b;c; 2025a;b; Wang et al., 2025b; Wang & Chen, 2025a;b; Zhao et al., 2024a; 2025b;a; 2024b; 2026; Li et al., 2025; Gao et al., 2024; 2025; Zhu et al., 2024; Lu et al., 2024b; Ren et al., 2025; Yu et al., 2025b). Instruction-based image editing has evolved rapidly. Early systems relied on modular, two-stage pipelines: a multimodal language model first produced textual prompts, spatial guidance, or synthetic instruction–image pairs, and a separate diffusion model then executed the edit—as in InstructEdit (Wang et al., 2023), InstructPix2Pix (Brooks et al., 2023), MagicBrush (Zhang et al., 2023b), and BrushEdit (Li et al., 2024c). Recent work has shifted toward tightly integrated, instruction-centric architectures. Models such as SmartEdit (Huang et al., 2024), X2I (Ma et al., 2025a), RPG (Yang et al., 2024), AnyEdit (Yu et al., 2025a), and UltraEdit (Zhao et al., 2024c) embed routing, task-aware objectives, and fine-grained controls directly into the network, yielding higher fidelity and more precise manipulation.

Unified generation-and-editing frameworks (e.g., OmniGen (Xiao et al., 2025), ACE (Han et al.), ACE++ (Mao et al., 2025), Lumina-OmniLV (Pu et al., 2025), Qwen2VL-Flux (Lu, 2024), DreamEngine (Chen et al., 2025), MetaQueries (Pan et al., 2025), Hidream-E1 (HiDream-ai, 2025)) treat editing as one capability of an end-to-end vision-language model, often fusing language embeddings with diffusion latents to provide context-aware, pixel-level control. Efficiency has advanced in parallel: ICEdit (Zhang et al., 2025) couples LoRA with mixture-of-experts tuning and optimized noise initialization, while SuperEdit (Ming et al., 2025) relies on higher-quality data and contrastive supervision to sustain performance at lower cost. Looking ahead, large foundation models such as Gemini (Gemini2.5, 2025) and GPT-5 (OpenAI, 2025) already show strong visual reasoning and coherent, instruction-guided image generation. Yet, despite extensive multimodal pre-training, they still fall short on image composition: object placement remains hard to control, lighting is often inconsistent, and subjects can drift in identity.

## A.3   SUBJECT-DRIVEN GENERATION

Extensive research has explored subject-driven image generation, in which the output must not only portray the contexts described by the text prompt but also faithfully include the specific subject supplied by reference images. Methods in this area are divided into two categories—test-time fine-tuning customization and zero-shot customization—according to whether extra training is needed for each new subject. **Our framework accommodates both categories, so we provide two corresponding variants in the main paper.**

**Test-time fine-tuning methods** (Gal et al., 2022; Ruiz et al., 2023) adapt a pre-trained T2I model to a small set of reference images (typically 3 to 5 images). Although this step adds computational cost and latency, it offers the greatest flexibility for diverse customization requirements. Such methods are commonly grouped into three subclasses: data regularization, weight regularization, and loss regularization. In the data-regularization family, DreamBooth (Ruiz et al., 2023) limits overfitting by generating superclass images with the base T2I model and training on both reference and regularization images; Custom Diffusion (Kumari et al., 2023) improves regularization quality by retrieving real images; and Specialist Diffusion (Lu et al., 2023a) applies extensive data augmentation. Weight-regularization approaches (Gal et al., 2022; Hu et al.; Han et al., 2023; Qiu et al., 2023) confine updates to carefully chosen parameters, such as the subject-specific text embeddings or the singular values of weight matrices. Loss-regularization approaches, including Specialist Diffusion (Lu et al., 2023a), MagiCapture (Hyung et al., 2024), and FaceChain-SuDe (Qiao et al., 2024), introduce objective terms that respectively maximize CLIP-space similarity between generated and reference images, disentangle identity and style via masked facial reconstruction, or encourage correct superclass classification.

**Zero-shot image customization methods** avoid subject-specific fine-tuning at inference time but rely on extensive pre-training. For general subject customization, InstantBooth (Shi et al., 2024a) adds a visual encoder that captures coarse-to-fine image features from the references; BLIP-Diffusion (Li et al., 2024a) fine-tunes BLIP-2 (Li et al., 2023a) to extract multimodal subject representations; ELITE (Wei et al., 2023) maps reference images into hierarchical textual tokens through global and local networks; and Song et al. (Song et al., 2025b) enhance textual control by removing the projection of visual embeddings onto textual embeddings. For facial customization, Instan-

tID (Wang et al., 2024b) isolates facial regions from reference images to extract appearance and structural cues. For style customization, InstantStyle (Wang et al., 2024a) identifies style-controlling layers and injects IP-Adapter features (Ye et al., 2023) into those layers to achieve style transfer. In-stantCharacter (Tao et al., 2025), IP-Adapter (Ye et al., 2023), and PuLID (Guo et al., 2024) have each released versions compatible with the FLUX model.

## B   VISUALIZING THE IMPACT OF ADAPTIVE BACKGROUND BLENDING

Although our loss function aims to find a new latent within the manifold learned by the adapter, encouraging the adapter-augmented T2I model's predictions to closely match those of the base model on the original noisy latent, this early-stage optimization primarily preserves structural elements rather than fine details. Consequently, discrepancies in fine-grained features often arise between the masked and unmasked regions. As illustrated in Fig. 9, we compare the composite images generated using our Adaptive Background Blending (ABB) method with those produced via direct background blending using the rectangular user mask. For clarity, we enlarge the boundary areas of each image (highlighted in pink dashed boxes) to better reveal differences.

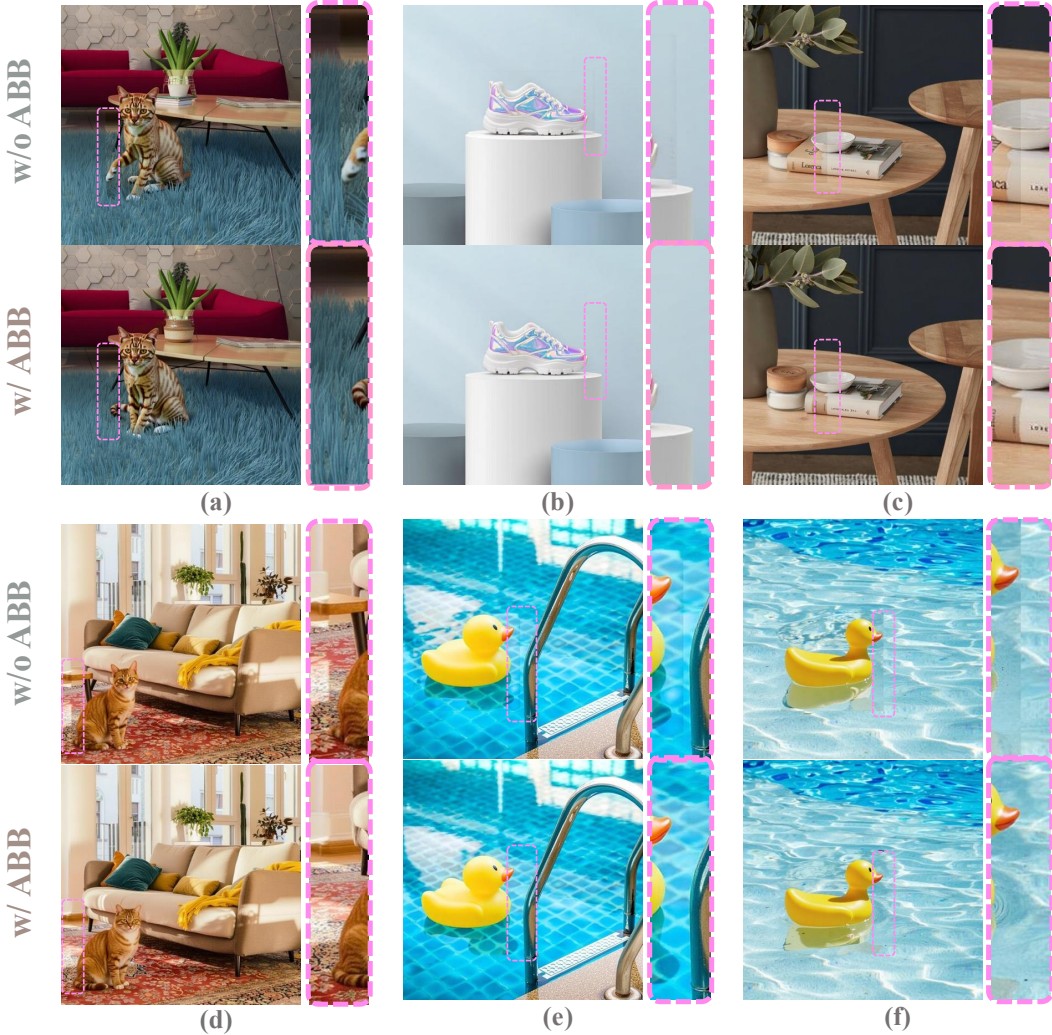

Figure 9: Comparison of composites from our Adaptive Background Blending (ABB) method and direct blending with a rectangular mask. Boundary regions within pink dashed boxes are enlarged for clarity. Please zoom in to see details.

## C    EQUIVALENCE OF QUERY BLURRING AND ATTENTION WEIGHT BLURRING

Consider a 2D Gaussian filter $G$ applied as a convolution operation, denoted by $\circledast$. The self-attention weights are computed as $QK^{\mathsf{T}}$, where $Q, K \in \mathbb{R}^{n \times d}$, with $n$ being the sequence length and $d$ the embedding dimension. We explore the effect of applying a Gaussian blur to the attention weights and its equivalence to blurring the query matrix.

### C.1    BLURRING THE QUERY MATRIX

Blurring the self-attention weights $QK^{\mathsf{T}}$ with a 2D Gaussian filter $G$ can be expressed as:

$$G \circledast (QK^{\mathsf{T}}), \tag{6}$$

where $\circledast$ denotes 2D convolution. Due to the linearity of convolution, there exists a Toeplitz matrix $B \in \mathbb{R}^{n \times n}$ such that the convolution operation can be represented as a matrix multiplication:

$$G \circledast (QK^{\mathsf{T}}) = B(QK^{\mathsf{T}}). \tag{7}$$

Using the properties of matrix multiplication, we can rewrite this as:

$$B(QK^{\mathsf{T}}) = (BQ)K^{\mathsf{T}}. \tag{8}$$

Since the convolution operation is linear, applying the Gaussian filter $G$ to the rows of $Q$ yields:

$$BQ = G \circledast Q. \tag{9}$$

Thus, we obtain:

$$G \circledast (QK^{\mathsf{T}}) = (G \circledast Q)K^{\mathsf{T}}. \tag{10}$$

This establishes that blurring the query matrix $Q$ with $G$ is mathematically equivalent to applying the same blur to the self-attention weights $QK^{\mathsf{T}}$. This equivalence suggests that query blurring can be used as a computationally efficient proxy for smoothing attention weights, potentially reducing the need for direct manipulation of the attention matrix.

### C.2    BLURRING THE KEY AND VALUE MATRICES

In contrast, applying the Gaussian blur to the key matrix $K$ does not yield a similar equivalence. Consider the convolution applied to $K$. The resulting attention weights become:

$$Q(G \circledast K)^{\mathsf{T}} = Q(BK)^{\mathsf{T}} = QK^{\mathsf{T}}B^{\mathsf{T}}. \tag{11}$$

Since $B^{\mathsf{T}} \neq B$ for a general Toeplitz matrix derived from a Gaussian filter, we have:

$$QK^{\mathsf{T}}B^{\mathsf{T}} \neq B(QK^{\mathsf{T}}). \tag{12}$$

Thus, blurring the key matrix $K$ does not produce an equivalent effect to blurring the attention weights $QK^{\mathsf{T}}$. A similar argument applies to the value matrix $V$, as the output of self-attention, $(QK^{\mathsf{T}})V$, involves $V$ in a post-multiplication step, and convolution on $V$ does not commute with the attention weight computation in the same manner.

### C.3    IMPLEMENTATION DETAILS OF GAUSSIAN BLURRING

For the 2D Gaussian filtering step, we adopt the implementation provided by the `kornia` library. Following standard engineering practice, we set the kernel radius to $r = 3\sigma$, since three standard deviations capture approximately $99.7\%$ of the Gaussian mass. Consequently, the kernel size is chosen as the nearest odd integer to $2r = 6\sigma$. In all of our experiments we use $\sigma = 10$ (see Tab. 5).

The procedure is applied to query embeddings by reshaping them into spatial feature maps, performing Gaussian smoothing, and then mapping them back into the sequence domain prior to attention computation. The full workflow is given below:

---

**Algorithm 2** Implementation Details of 2D Gaussian Blurring

---

**Input:** A query matrix $\mathbf{Q} \in \mathbb{R}^{B \times L \times D}$, key $\mathbf{K}$, value $\mathbf{V}$, spatial dimensions $(H, W)$, Gaussian standard deviation $\sigma$.
**Output:** Attention output $\mathbf{O}$.

---

```
1: // Reshape query into spatial tensor
```
2: $\mathbf{Q}_{sp} \leftarrow \text{Reshape}(\mathbf{Q}, (B, H, W, D))$
3: $\mathbf{Q}_{sp} \leftarrow \text{Permute}(\mathbf{Q}_{sp}, (0, 3, 1, 2))$           $\triangleright B \times D \times H \times W$
```
4: // Construct Gaussian kernel size
```
5: $k \leftarrow 6\sigma$
6: $k \leftarrow k - (k \bmod 2) + 1$           $\triangleright$ Ensure odd kernel size
7: $\text{kernel\_size} \leftarrow (k, k)$
8: $\boldsymbol{\sigma} \leftarrow (\sigma, \sigma)$
```
9: // Apply 2D Gaussian smoothing
```
10: $\mathbf{Q}_{sp} \leftarrow \text{kornia.filters.gaussian\_blur2d}(\mathbf{Q}_{sp}, \text{kernel\_size}, \boldsymbol{\sigma})$
```
11: // Reshape smoothed queries back to sequence form
```
12: $\mathbf{Q}' \leftarrow \text{Permute}(\mathbf{Q}_{sp}, (0, 2, 3, 1))$
13: $\mathbf{Q}' \leftarrow \text{Reshape}(\mathbf{Q}', (B, L, D))$
```
14: // Compute attention
```
15: $\mathbf{A} \leftarrow \text{softmax}\left(\frac{\mathbf{Q}'\mathbf{K}^\top}{\sqrt{D}}\right)$
16: $\mathbf{O} \leftarrow \mathbf{A}\mathbf{V}$
17: **return** $\mathbf{O}$

---

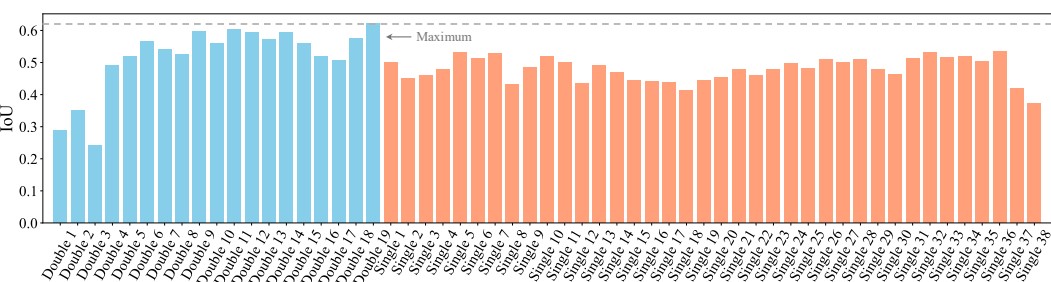

Figure 10: The IoU is calculated between the mask produced from each block and the ground-truth mask, which is obtained by segmenting the final generated images using SAM. The IoU for each block is averaged over 100 images.

## D  EVALUATING CROSS-ATTENTION MAP ACCURACY VIA IoU

To identify the most accurate cross-attention maps that reflect the location of the generated object, we first create 100 prompts containing a main subject (e.g.,"a dog is sleeping on a couch") using GPT-5. These prompts are then used to generate 100 images with FLUX.1-dev, employing 20 denoising steps. Cross-attention maps are extracted from 19 multi-stream (or double-stream) blocks and 38 single-stream blocks across all denoising steps. The maps are averaged over the 20 steps and subsequently normalized and binarized, resulting in a total of 57 binary masks.

To determine which of these 57 masks is the most accurate, we compute the Intersection over Union (IoU) between each mask and a ground-truth mask obtained by segmenting the final generated images using SAM (Ravi et al., 2025). The IoU for each block is averaged over the 100 generated images. The results are presented in Fig. 10, showing that the cross-attention maps from the last multi-stream (or double-stream) block achieve the highest segmentation accuracy. A visualization of the cross-attention maps from all blocks is provided in Fig. 11.

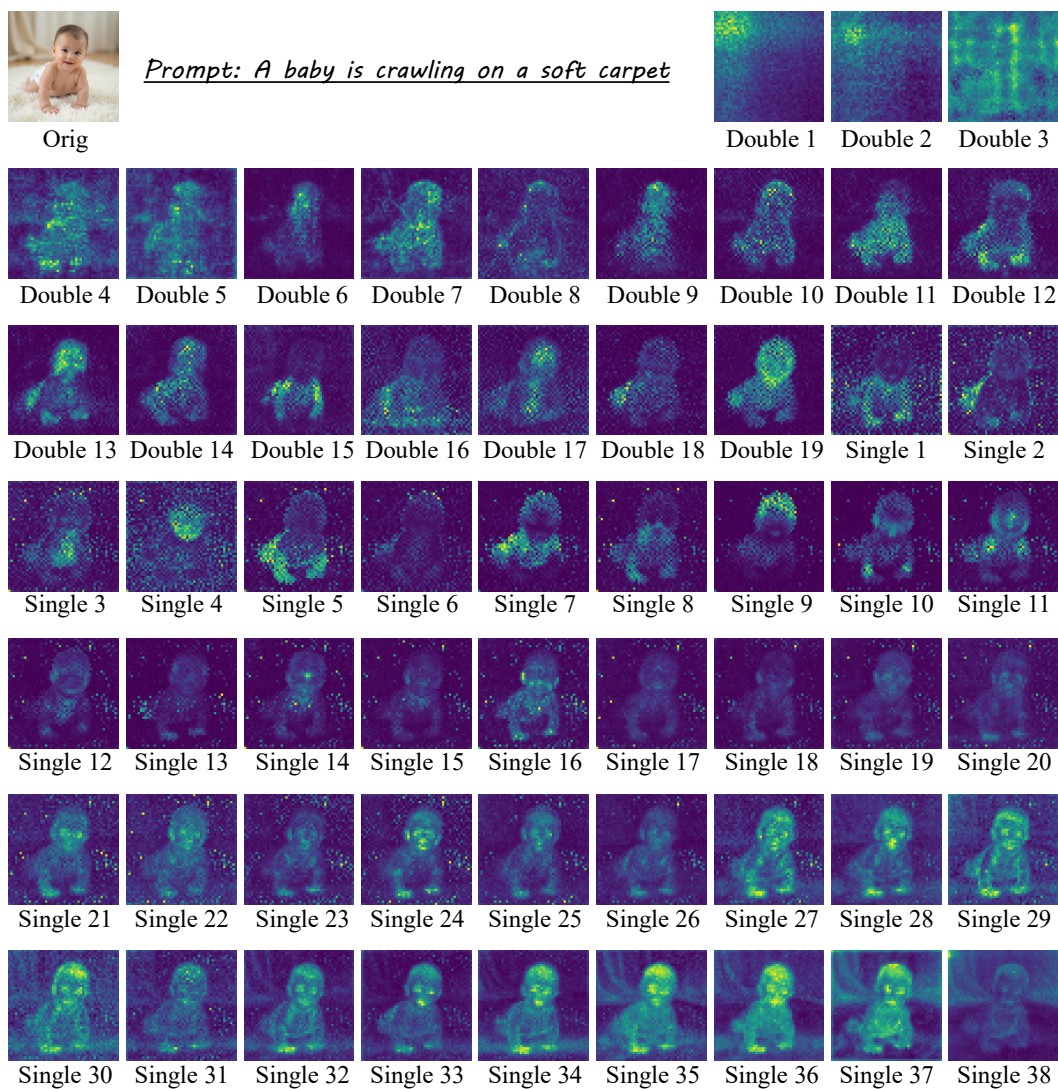

Figure 11: Visualization of cross-attention maps from different MMDiT blocks of FLUX.1-dev.

# E EXPERIMENTS WITH SDXL, SD3.5, AND PIXART

Our framework introduces a novel, model-agnostic approach for enhancing generative architectures, anchored by three synergistic components: Manifold-Steered Anchor (MSA) loss, Degradation-Suppression Guidance (DSG), and Adaptive Background Blending (ABB). These components are meticulously designed to leverage ubiquitous features of modern generative models, ensuring seamless integration without requiring architectural modifications. Specifically, MSA utilizes either LoRA-based personalization or a pretrained personalization adapter, DSG capitalizes on widely available self-attention maps, and ABB harnesses text–image cross-attention maps, a staple in most text-to-image pipelines. This design ensures broad applicability across diverse generative models.

In the main text, we showcase the effectiveness of our approach on Flux, a leading open-source model. To establish its generalizability, we conduct experiments on SDXL (Podell et al., 2024), SD3.5 (Esser et al., 2024), and PixArt (Chen et al.) across DreamEditBench and ComplexCompo. The results are presented in Tab. 3. On DreamEditBench, our LoRA-based variant with Flux achieves state-of-the-art performance in subject identity preservation, while delivering superior image quality. Similarly, on ComplexCompo, the Flux-LoRA configuration excels in identity consistency and image fidelity. Notably, the framework's benefits extend beyond a single model family: both SDXL and

PixArt-$\Sigma$ exhibit substantial performance gains, affirming the approach's generality and adaptability across diverse generative architectures.

Table 3: Comparison of compositional performance across two benchmarks with **different base models**. The best result in each column is highlighted in **bold**, while the second-best is underlined. Metrics shown in pink are those specifically trained to better align with human preferences. Abbreviations: IRF: Instance Retrieval Features; IR = ImageReward; VR = VisionReward.

| Bench | Method | Base Model | Subject Identity Consistency | | | | Background | | Image Quality | |
|---|---|---|---|---|---|---|---|---|---|---|
| | | | CLIP-I ↑ | DINOv2 ↑ | IRF ↑ | DreamSim ↓ | LPIPS ↓ | SSIM ↑ | IR ↑ | VR ↑ |
| DreamEdit-Bench (220) | Flux.1 Fill | FLUX | 0.7328 | 0.6745 | 0.5754 | 0.5233 | 0.0166 | 0.9076 | 0.5577 | 3.5997 |
| | Ours-Adapter | SDXL | 0.7944 | 0.7334 | 0.7659 | 0.3761 | 0.0238 | 0.8922 | 0.5621 | 3.6158 |
| | Ours-Adapter | SD3.5 | 0.8054 | 0.7407 | 0.7699 | 0.3745 | **0.0234** | 0.8937 | 0.5701 | 3.6187 |
| | Ours-LoRA | PixArt-$\Sigma$ | 0.8098 | 0.7445 | 0.7798 | 0.3612 | 0.0251 | 0.8875 | 0.5842 | 3.6198 |
| | Ours-Adapter | FLUX | 0.8086 | 0.7415 | 0.7702 | 0.3730 | 0.0236 | **0.8959** | 0.5709 | **3.6234** |
| | Ours-LoRA | FLUX | **0.8125** | **0.7452** | **0.7900** | **0.3577** | 0.0271 | 0.8847 | **0.5906** | 3.6161 |
| Complex-Compo (300) | Flux.1 Fill | FLUX | 0.7108 | 0.6475 | 0.5466 | 0.6018 | 0.0232 | 0.7442 | 0.4088 | 3.5737 |
| | Ours-Adapter | SDXL | 0.7657 | 0.7084 | 0.6862 | 0.4457 | 0.0457 | 0.7612 | 0.3894 | 3.5987 |
| | Ours-Adapter | SD3.5 | 0.7701 | 0.7091 | 0.6977 | 0.4173 | **0.0401** | 0.7784 | 0.4091 | **3.6021** |
| | Ours-LoRA | PixArt-$\Sigma$ | 0.7924 | 0.7287 | 0.7311 | 0.3603 | 0.0424 | 0.7698 | **0.4277** | 3.5988 |
| | Ours-Adapter | FLUX | 0.7721 | 0.7107 | 0.6764 | 0.4294 | 0.0404 | **0.7789** | 0.4090 | 3.6020 |
| | Ours-LoRA | FLUX | **0.7999** | **0.7384** | **0.7659** | **0.3542** | 0.0430 | 0.7634 | 0.4246 | 3.5951 |

# F  USER STUDY

We conduct a user study involving 50 participants. Each participant was asked to complete 50 ranking tasks. In each task, they were shown 13 composition results generated by different methods, along with a reference subject image.

To ensure a balanced evaluation, 25 of the tasks were randomly sampled from DreamEditBench and the remaining 25 from ComplexCompo. Participants were asked to rank the results based on two key criteria: (1) subject identity consistency and (2) composition realism. A lower rank (e.g., 1st) indicates a better composition result, while a higher rank (e.g., 13th) reflects a less favorable outcome.

We summarize the average ranking scores for each method in Tab. 4. Our method received the most favorable rankings from the majority of participants, demonstrating its effectiveness in producing high-quality compositions.

# G  BENCHMARK DETAILS

Our benchmark consists of 300 triplets, each comprising a subject image, a background image, and a bounding box. The subject images are identical to those used in DreamEditBench (Li et al., 2023b; Ruiz et al., 2023), while the background images are sampled from the OpenImages dataset (Kuznetsova et al., 2020). These backgrounds exhibit a variety of aspect ratios and resolutions, including landscape and portrait formats, such as $768 \times 1088$, $768 \times 1072$, $768 \times 1024$, $768 \times 1152$, $1024 \times 768$, $1152 \times 768$, $1200 \times 768$, $848 \times 768$, and $1360 \times 768$. The bounding boxes are manually designed to ensure that the size and placement of the inserted subjects are contextually appropriate and visually plausible. The benchmark will be released publicly.

Table 4: Average ranking scores from the user study on image composition methods. Lower is better.

| Method | Training-Free | Average Ranking (Lower is Better) |
|---|:---:|:---:|
| MADD | ✗ | 12.44 |
| ObjectStitch | ✗ | 11.80 |
| DreamCom | ✗ | 6.66 |
| AnyDoor | ✗ | 4.12 |
| UniCombine | ✗ | 2.94 |
| PBE | ✗ | 4.94 |
| TIGIC | ✔ | 9.74 |
| TALE | ✔ | 9.06 |
| TF-ICON | ✔ | 8.36 |
| DreamEdit | ✔ | 6.36 |
| EEdit | ✔ | 10.76 |
| Ours-Adapter | ✔ | 2.30 |
| Ours-LoRA | ✔ | **1.52** |

## H   PROMPTS FOR PROPRIETARY FOUNDATION MODELS

To perform image composition with proprietary, general-purpose multimodal foundation models (e.g., GPT-5 (OpenAI, 2025), Gemini 2.5 Pro (Gemini2.5, 2025), SeedEdit/Doubao (Shi et al., 2024b), and Grok 4 (gro, 2025)), we upload three images: (1) Subject image; (2) Background image; and (3) Mask image defining the insertion region.

We then issue a prompt of the following form (with the resolution and coordinates adjusted for each case):

> *Please insert the object from the first uploaded image into the second image. The target region for insertion is defined by the mask in the third image. For reference, the resolution of the second image is $1152 \times 768$, and the bounding box for placement is specified by the top-left and bottom-right coordinates: $(x_1 = 550, y_1 = 600, x_2 = 700, y_2 = 750)$. The inserted object should retain the same identity and appearance as in the first image. The final composite should appear realistic, natural, and physically plausible.*

## I   SUBJECT IDENTITY METRICS ANALYSIS

In our experiments we found that widely-used subject-identity metrics such as CLIP-I (Radford et al., 2021) and DINOv2 (Oquab et al., 2024) correlate poorly with human preferences. Because they focus almost exclusively on semantic similarity, they ignore appearance changes introduced by lighting, shadows, reflections, and surrounding context. Fig. 12(b) presents several image pairs produced by AnyDoor (left) and by our method (right); the corresponding CLIP-I ($\uparrow$) (Radford et al., 2021), DINOv2 ($\uparrow$) (Oquab et al., 2024), IRF ($\uparrow$) (Shao & Cui, 2022), and DreamSim ($\downarrow$) (Fu et al., 2023) scores are shown beneath each image, with the better score highlighted in red. Although the AnyDoor results are visibly less realistic and less consistent, they nevertheless receive higher CLIP-I and DINOv2 scores, and in most cases higher IRF scores, demonstrating that these measures do not faithfully capture compositional quality.

A reliable metric should recognise the same object whether it is underwater (see Fig. 12(b)[(8), (17), (20)]), in shadow (see Fig. 12(b)[(9), (13), (19)]), partially occluded (see Fig. 12(b)[(2)]), or situated in a low-light or back-lit scene (see Fig. 12(b)[(5), (14), (15), (16)]). Among the metrics we evaluated, only DreamSim, which was designed to align more closely with human perception, consistently exhibits this desired behaviour.

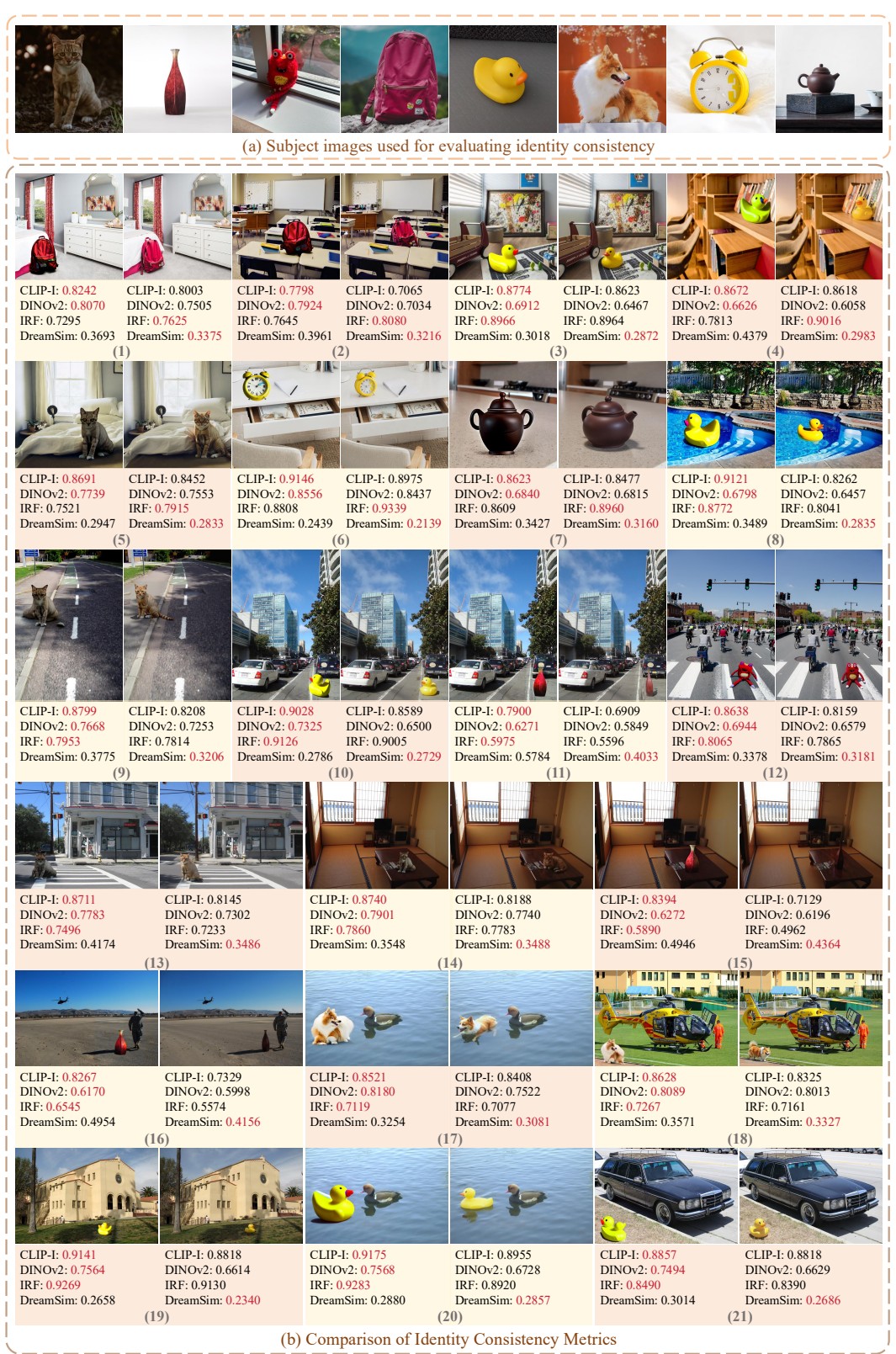

Figure 12: Comparison of Subject Identity Metrics. (a) Reference subject images used for metric calculations. (b) Image pairs generated by AnyDoor (left) and our method (right), with corresponding CLIP-I (↑), DINOv2 (↑), IRF (↑), and DreamSim (↓) scores displayed below each image; the better score is highlighted in red. Despite AnyDoor's results appearing less realistic and consistent, they often achieve higher CLIP-I, DINOv2, and IRF scores, indicating that these metrics may not reliably reflect compositional quality. In contrast, DreamSim provides a more reliable assessment.

Table 5: Hyperparameters of Our Frameworks. Bin Thresh = Binarization Threshold; #iter = Number of Iterations.

| Variant | Denoising Setup | | Manifold-Steered Anchor Loss | | | Degradation-Suppression Guidance | | | Adaptive Background Blending | | |
| | Total steps $(T-1) \to 0$ | Start step $t_1$ | Step Range $t_1 \to \tau$ | Learning Rate $\alpha$ | #iters $k$ | Step Range $t_1 \to 0$ | Scale $\eta$ | Blur $\sigma$ | Step Range $(t_1-1) \to 1$ | Bin Thresh | Dilation Kernel Size |
|---|---|---|---|---|---|---|---|---|---|---|---|
| Ours-Adapter | $19 \to 0$ | 14 | $14 \to 12$ | 500, 750, 1000 | 10 | $14 \to 0$ | 0.5 | 10 | $13 \to 1$ | 0.2 | 3 |
| Ours-LoRA | $19 \to 0$ | 13 | $13 \to 12$ | 50, 300 | 2 | $13 \to 0$ | 0.7 | 10 | $12 \to 1$ | 0.4 | 3 |

## J IMPLEMENTATION DETAILS

The hyperparameters used in our frameworks are summarized in Tab. 5. Under Denoising Setup, "Total steps" refers to the full diffusion/noising schedule, which specifies a sequence of 20 values of $\sigma_t$ across timesteps $t$. However, our method does not begin denoising at the first timestep. As shown in Algorithm 1, it starts at $t_1 = 14$ (Ours-Adapter), resulting in 15 denoising steps in total. Under MSA loss, "Step Range" indicates the subset of denoising steps to which MSA optimization is applied. For the adapter setting, the MSA loss is applied only to the first three denoising steps (from $t = 14$ to $t = 12$). In addition, since a LoRA is trained for a specific subject, it provides a more tailored prior than the generic adapter, allowing it to converge in fewer steps and with lower overall compute.

Each baseline is implemented according to the configuration settings recommended in its original publication. The repositories utilized for each baseline are listed below:

1. MADD: https://github.com/KaKituken/affordance-aware-any
2. ObjectStitch: https://github.com/bcmi/ObjectStitch-Image-Composition
3. DreamCom: https://github.com/bcmi/DreamCom-Image-Composition
4. AnyDoor: https://github.com/ali-vilab/AnyDoor
5. UniCombine: https://github.com/Xuan-World/UniCombine
6. PBE: https://github.com/Fantasy-Studio/Paint-by-Example
7. TIGIC: https://github.com/zrealli/TIGIC
8. TALE: https://github.com/tkpham3105/TALE
9. TF-ICON: https://github.com/Shilin-LU/TF-ICON
10. DreamEdit: https://github.com/DreamEditBenchTeam/DreamEdit
11. EEdit: https://github.com/yuriYanZeXuan/EEdit

## K DISCUSSION ON INVERSION VS. ONE-STEP FORWARD DIFFUSION

We provide an expanded discussion of our design choice between inversion and one-step forward diffusion to better clarify the motivation behind our approach. In training-free image editing, both inversion and one-step forward diffusion are commonly used to obtain a noisy latent that serves as the starting point for subsequent optimization or denoising. In our framework, we intentionally adopt one-step forward diffusion as a practical substitute for inversion. Our method does not depend on the initial latent to preserve object identity. Instead, the MSA loss extracts object-specific information from the adapter/LoRA and injects it into the latent during the editing process.

This design choice is motivated by a practical observation: many modern models are distilled for speed, making accurate inversion difficult to achieve in practice. When inversion cannot reliably encode object identity, its benefit becomes limited. In such cases, a noisy latent that still retains enough background structure is sufficient as a starting point. For this reason, we adopt one-step forward diffusion as a pragmatic replacement for inversion, rather than a theoretically equivalent alternative. It offers two advantages: (i) it avoids the accuracy limitations of inversion on distilled models, making it more broadly applicable, and (ii) it is faster than performing an inversion.

Table 6: Image Quality Evaluation Results using HPSv3 and UnifiedReward variants.

| Bench | Method | HPSv3 | UnifiedReward-2.0-qwen3vl-8b | | | | | UnifiedReward-Edit-qwen3vl-8b | | | | | UnifiedReward-Think-qwen-7b | | | | |
|---|---|---|---|---|---|---|---|---|---|---|---|---|---|---|---|---|---|
| | | | Instruction | | Quality | | Average | Instruction | | Quality | | Average | Instruction | | Quality | | Average |
| | | | Success | Overedit | Natural | Artifact | | Success | Overedit | Natural | Artifact | | Success | Overedit | Natural | Artifact | |
| Dream-Edit-Bench (220) | MADD (He et al., 2024) | 1.2443 | 14.2273 | 12.3727 | 16.6153 | 18.6807 | 15.4740 | 11.8091 | 17.8727 | 13.1045 | 12.4727 | 13.8148 | 14.7553 | 16.2128 | 19.1103 | 20.5724 | 17.6627 |
| | ObjectStitch (Song et al., 2023) | 7.4529 | 19.9954 | 16.8721 | 21.0455 | 21.8864 | 19.9499 | 16.2500 | 21.2000 | 18.6545 | 20.6500 | 19.1886 | 20.7500 | 17.7614 | 21.0563 | 20.7676 | 20.0838 |
| | DreamCom (Lu et al., 2023c) | 5.9324 | 18.1818 | 15.4818 | 23.5388 | 24.0502 | 20.3132 | 14.0727 | 22.5409 | 21.0091 | 22.0955 | 19.9296 | 18.5263 | 15.4737 | 21.6641 | 21.6953 | 19.3399 |
| | AnyDoor (Chen et al., 2024c) | 8.4867 | 23.2146 | 17.8128 | 20.9772 | 21.9361 | 20.9852 | 19.0909 | 20.6000 | 17.4182 | 19.2864 | 19.0989 | 21.2083 | 18.5694 | 21.9214 | 21.5929 | 20.8230 |
| | UniCombine (Wang et al., 2025a) | 8.8415 | 22.4545 | 17.8000 | 23.1682 | 23.9000 | 21.8307 | 20.1591 | 22.7773 | 21.2500 | 22.6455 | 21.7080 | 23.0595 | 16.2381 | 21.9520 | 20.8400 | 20.5224 |
| | PBE (Yang et al., 2023) | 8.3789 | 22.4292 | 17.5205 | 22.2773 | 23.2273 | 21.3636 | 18.2318 | 21.8182 | 19.5955 | 21.2091 | 20.2137 | 24.1481 | 17.8025 | 21.5786 | 21.9500 | 21.3698 |
| | TIGIC (Li et al., 2024b) | 5.2676 | 17.3136 | 14.8045 | 19.0318 | 20.3636 | 17.8784 | 13.7045 | 19.4636 | 17.0864 | 18.1455 | 17.1000 | 17.7294 | 17.0000 | 21.0897 | 20.8828 | 19.1755 |
| | TALE (Pham et al., 2024) | 6.3773 | 19.6027 | 15.9863 | 20.4455 | 21.4773 | 19.3780 | 14.8318 | 21.6182 | 17.3227 | 18.5409 | 18.0784 | 21.6071 | 17.6548 | 21.0667 | 20.6933 | 20.2555 |
| | TF-ICON (Lu et al., 2023d) | 7.2643 | 20.4490 | 16.7538 | 20.8273 | 21.7227 | 19.9382 | 15.9045 | 20.1727 | 17.9909 | 19.0182 | 18.2716 | 20.9870 | 16.9870 | 21.2946 | 20.7984 | 20.0168 |
| | DreamEdit (Li et al., 2023b) | 6.0250 | 19.9227 | 16.5409 | 19.2773 | 20.5227 | 19.0659 | 14.9273 | 19.1818 | 13.3909 | 15.5545 | 15.7636 | 20.4714 | 16.8143 | 20.3333 | 20.7153 | 19.5836 |
| | EEdit (Yan et al., 2025) | 6.6689 | 18.3790 | 15.3379 | 22.0636 | 23.4818 | 19.8156 | 14.2091 | 22.2864 | 19.9500 | 21.7955 | 19.5603 | 21.1463 | 18.2561 | 22.0635 | 21.2222 | 20.6720 |
| | Ours-Adapter | 8.8861 | 23.3727 | 17.0500 | 23.5727 | 23.8136 | 21.9523 | 21.3364 | 22.7591 | 21.3636 | 22.6136 | 22.0182 | 23.2222 | 18.7407 | 22.8448 | 22.0086 | 21.7041 |
| | Ours-LoRA | 8.8688 | 23.4545 | 16.8136 | 23.7727 | 23.8500 | 21.9727 | 21.1273 | 22.7455 | 21.3000 | 22.5955 | 21.9421 | 23.7590 | 18.5904 | 22.4853 | 20.6765 | 21.3778 |
| Complex-Compo (300) | MADD (He et al., 2024) | 5.9673 | 13.8900 | 11.8167 | 18.8633 | 17.1333 | 15.4258 | 12.6800 | 16.7300 | 12.5167 | 10.3000 | 13.0567 | 17.6538 | 15.7115 | 20.2867 | 19.5533 | 17.2524 |
| | ObjectStitch (Song et al., 2023) | 8.8389 | 20.7157 | 13.1773 | 21.4200 | 19.4800 | 18.6983 | 17.3567 | 21.8500 | 17.1333 | 18.9733 | 18.8283 | 19.8304 | 14.6518 | 21.3467 | 19.5400 | 18.8394 |
| | DreamCom (Lu et al., 2023c) | 7.9884 | 8.2234 | 9.0756 | 23.9178 | 23.4737 | 16.1726 | 5.9507 | 23.7072 | 22.3618 | 22.4375 | 18.6143 | 17.8810 | 15.2857 | 22.5000 | 20.9737 | 19.0509 |
| | AnyDoor (Chen et al., 2024c) | 8.9760 | 21.7633 | 13.1133 | 20.5567 | 18.5300 | 18.4908 | 18.4967 | 21.8300 | 14.9267 | 18.1667 | 18.3550 | 22.6606 | 17.6422 | 21.2467 | 19.5333 | 19.8876 |
| | UniCombine (Wang et al., 2025a) | 8.8999 | 15.6747 | 11.2226 | 23.0878 | 22.7230 | 18.1770 | 13.7399 | 23.4966 | 20.7195 | 21.4554 | 19.8529 | 20.3646 | 15.7500 | 21.9764 | 20.2804 | 19.6449 |
| | PBE (Yang et al., 2023) | 8.5923 | 19.9151 | 13.1283 | 21.9243 | 20.1349 | 18.7007 | 16.8947 | 21.9605 | 17.3257 | 19.6217 | 18.9507 | 23.5810 | 14.7048 | 21.2039 | 19.6447 | 19.6170 |
| | TIGIC (Li et al., 2024b) | 7.6630 | 14.6250 | 11.9899 | 21.2357 | 20.0202 | 16.9677 | 12.6027 | 19.5185 | 16.9192 | 16.6801 | 16.4301 | 16.7168 | 16.5398 | 21.5051 | 20.2862 | 18.2956 |
| | TALE (Pham et al., 2024) | 8.7351 | 16.9899 | 11.3826 | 22.7600 | 20.7000 | 17.9581 | 15.0100 | 22.3567 | 19.0333 | 18.6267 | 18.7567 | 17.8317 | 15.4455 | 21.9667 | 19.9767 | 18.7955 |
| | TF-ICON (Lu et al., 2023d) | 9.3258 | 17.7047 | 12.6812 | 21.9463 | 20.4799 | 18.2030 | 15.1477 | 21.2919 | 18.1577 | 18.3490 | 18.2366 | 19.9775 | 16.7072 | 21.5101 | 19.7584 | 19.2380 |
| | DreamEdit (Li et al., 2023b) | 8.0434 | 17.2600 | 12.0233 | 18.5669 | 17.2578 | 16.2770 | 14.7300 | 19.4600 | 13.7467 | 15.6367 | 15.8934 | 23.3964 | 15.6396 | 20.4333 | 19.5400 | 18.9805 |
| | EEdit (Yan et al., 2025) | 8.7835 | 14.9500 | 11.2567 | 22.8746 | 21.8152 | 17.7241 | 13.4224 | 23.1485 | 20.2601 | 22.1081 | 19.7348 | 22.8058 | 15.6990 | 22.3498 | 20.5941 | 20.2367 |
| | Ours-Adapter | 9.6485 | 22.6162 | 14.2525 | 22.5552 | 21.9064 | 20.3326 | 20.4582 | 22.6421 | 18.5518 | 21.2876 | 20.7349 | 24.0300 | 17.4600 | 22.2074 | 20.3913 | 20.9647 |
| | Ours-LoRA | 9.8418 | 22.9532 | 13.3779 | 22.9130 | 21.7525 | 20.2492 | 20.8328 | 22.8261 | 19.1940 | 21.2776 | 21.0326 | 24.3400 | 17.8900 | 22.3838 | 19.9596 | 21.1212 |

EEdit also proposes an elegant strategy, termed inversion skipping, to accelerate the initialization process. In EEdit, the initial latent is obtained via an inversion procedure involving model predictions, while inversion skipping significantly reduces the number of required steps. In contrast, our one-step forward diffusion does not perform inversion at all. The initial latent is produced by directly sampling noise and adding it to the clean latent at a chosen timestep, without any model prediction. This makes our initialization computationally lighter. Because FLUX-Dev is a CFG-distilled model, its inversions are relatively imprecise, which we believe contributes to the weaker subject-identity preservation observed in EEdit. Our method is therefore designed to avoid reliance on accurate inversion in such distilled settings.

# L  ADDITIONAL IMAGE QUALITY EVALUATION USING UNIFIEDREWARD AND HPSV3

To provide a more comprehensive assessment of composition quality, we further evaluate the methods using three variants of UnifiedReward (Wang et al., 2025c;d) (UnifiedReward-2.0-qwen3vl-8b, UnifiedReward-Edit-qwen3vl-8b, and UnifiedReward-Think-qwen-7b), as well as HPSv3 (Ma et al., 2025b). The results are summarized in Tab. 6.

# M  FURTHER ANALYSIS ON DSG IN SD3.5

We conduct experiments on SD3.5 by replacing our DSG mechanism with standard negative prompting. Specifically, in Eqn. 3, we substitute the negative velocity $v_{\theta+\Delta\theta}^{\text{neg}}$ with the velocity obtained using a variety of commonly used negative prompts. The four sets of negative prompts used are:

1. distorted, deformed, glitch, artifacts

2. undefined shapes, bad anatomy, unnatural pose

3. low quality, worst quality, low resolution, blurry, out of focus

4. AI artifacts, melted objects, strange textures

The quantitative results on the DreamEditBench dataset are presented in Tab. 7.

**Key Insights**  These results lead to the following key conclusions:

1. **Performance Sensitivity:** The performance of standard negative prompting is sensitive to the specific wording used (e.g., Prompt 2 performs better than Prompt 4). This confirms that the effectiveness of heuristic negative prompts relies heavily on manual, often tedious, prompt engineering.

Table 7: Comparison of negative prompting versus DSG on SD3.5 on the DreamEditBench.

| Method | CLIP ↑ | DINO ↑ | IRF ↑ | DreamSim ↓ | LPIPS ↓ | SSIM ↑ | IR ↓ | VR ↑ |
|---|---|---|---|---|---|---|---|---|
| Ours-SD3.5-Adapter (w/ NP 1) | 0.7934 | 0.7311 | 0.7608 | 0.3918 | 0.0331 | 0.8812 | 0.5837 | 3.5714 |
| Ours-SD3.5-Adapter (w/ NP 2) | 0.8029 | 0.7388 | 0.7679 | 0.3781 | 0.0259 | 0.8908 | 0.5726 | 3.6102 |
| Ours-SD3.5-Adapter (w/ NP 3) | 0.7958 | 0.7329 | 0.7623 | 0.3892 | 0.0315 | 0.8839 | 0.5814 | 3.5796 |
| Ours-SD3.5-Adapter (w/ NP 4) | 0.7887 | 0.7281 | 0.7574 | 0.3976 | 0.0368 | 0.8765 | 0.5882 | 3.5548 |
| **Ours-SD3.5-Adapter (w/ DSG)** | **0.8054** | **0.7407** | **0.7699** | **0.3745** | **0.0234** | **0.8937** | **0.5701** | **3.6187** |

2. **Adaptive Guidance:** Standard negative prompts generate a degradation direction that is decoupled from the specifics of the input image. In contrast, DSG constructs an image-specific low-quality direction by blurring the attention component, making it significantly more adaptive, stable, and effective across diverse inputs without requiring any manual prompt tuning. DSG consistently outperforms all tested standard negative prompts across all metrics.

This analysis confirms that DSG provides a superior, more robust, and automated mechanism for degradation suppression compared to traditional negative prompting, even in models like SD3.5 where negative prompts are generally effective.

# N    RUNTIME COMPARISON

The wall-clock runtime at a $512 \times 512$ resolution on an H100 GPU is summarized in the table below. In our implementation, we applied qint8 quantization to all FLUX-based methods to accelerate inference and reduce GPU memory usage. Additionally, Ours-Adapter can also be run on a 24-GB GPU by enabling CPU offloading.

Table 8: Runtime and memory usage comparison of various image composition methods at a $512 \times 512$ resolution.

| Method | Training Free | Base Model | External Model | Time (s) | Peak Memory (MB) |
|---|---|---|---|---|---|
| MADD (He et al., 2024) | ✗ | SD | DINO | 45.73 | 11708 |
| ObjectStitch (Song et al., 2023) | ✗ | SD | VIT | 6.63 | 8268 |
| DreamCom (Lu et al., 2023c) | ✗ | SD | LoRA | 9.87 | 3388 |
| AnyDoor (Chen et al., 2024c) | ✗ | SD | DINO | 8.61 | 18612 |
| UniCombine (Wang et al., 2025a) | ✗ | FLUX | LoRA | 11.98 | 22711 |
| PBE (Yang et al., 2023) | ✗ | SD | - | 3.52 | 10842 |
| TIGIC (Li et al., 2024b) | ✔ | SD | - | 10.82 | 21640 |
| TALE (Pham et al., 2024) | ✔ | SD | - | 8.03 | 23524 |
| TF-ICON (Lu et al., 2023d) | ✔ | SD | - | 24.55 | 20670 |
| DreamEdit (Li et al., 2023b) | ✔ | SD | LoRA, VIT | 99.83 | 19298 |
| EEdit (Yan et al., 2025) | ✔ | FLUX | - | 60.31 | 26546 |
| Ours-Adapter | ✔ | FLUX | Adapter | 38.29 | 32552 |
| Ours-LoRA | ✔ | FLUX | LoRA | 18.08 | 23519 |

# O    ADDITIONAL QUALITATIVE RESULTS

We offer more qualitative assessment results, including visualizations of all baselines, presented in Figs. 13 to 18.

# P    LLM USAGE STATEMENT

We used large language models for text polishing and grammar correction during manuscript preparation. No LLMs were involved in the design of the method, experiments, or analysis. All content has been carefully verified and validated by the authors.

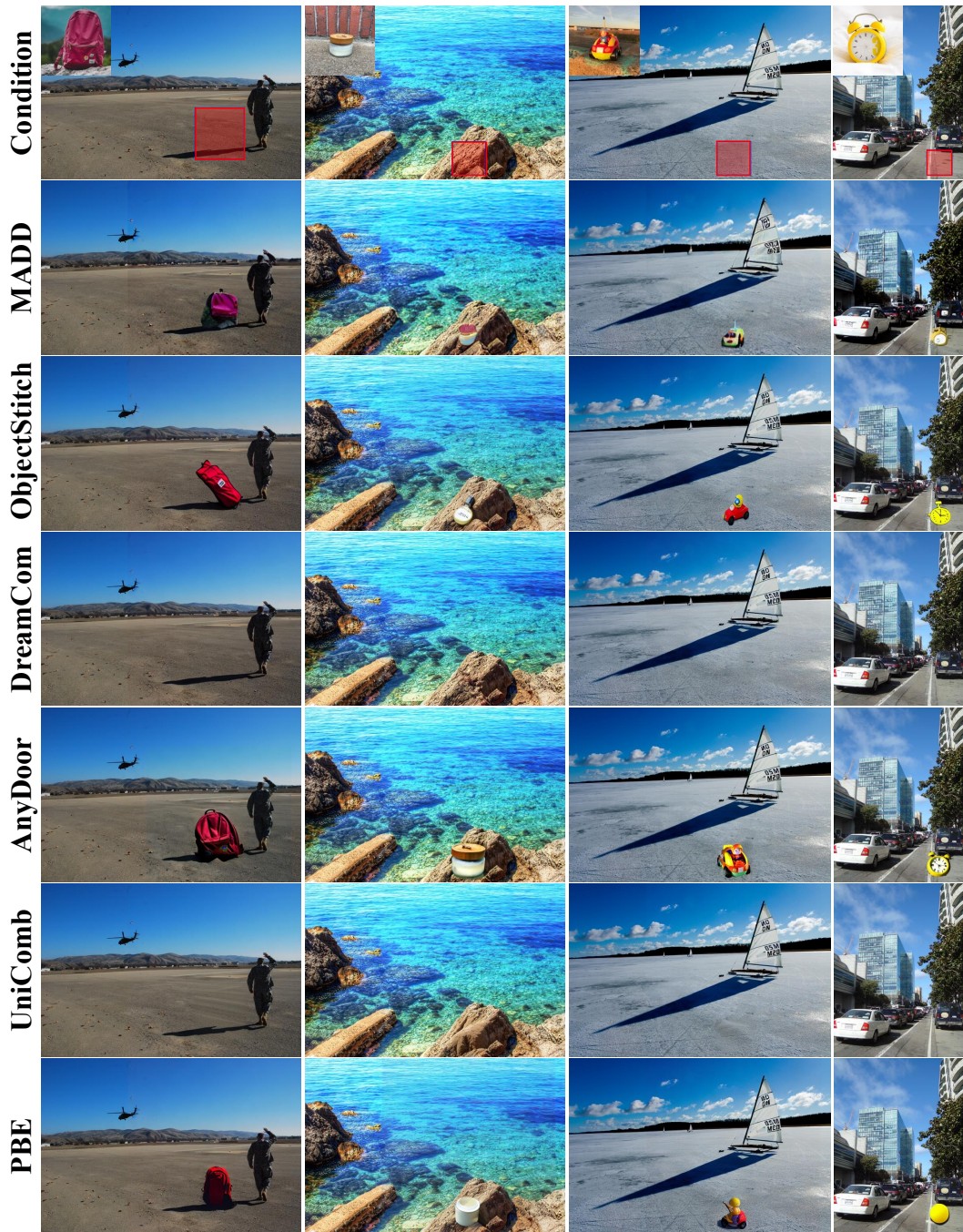

Figure 13: **(Part 1 of 2)** Qualitative comparison of our method against baselines in challenging scenarios.

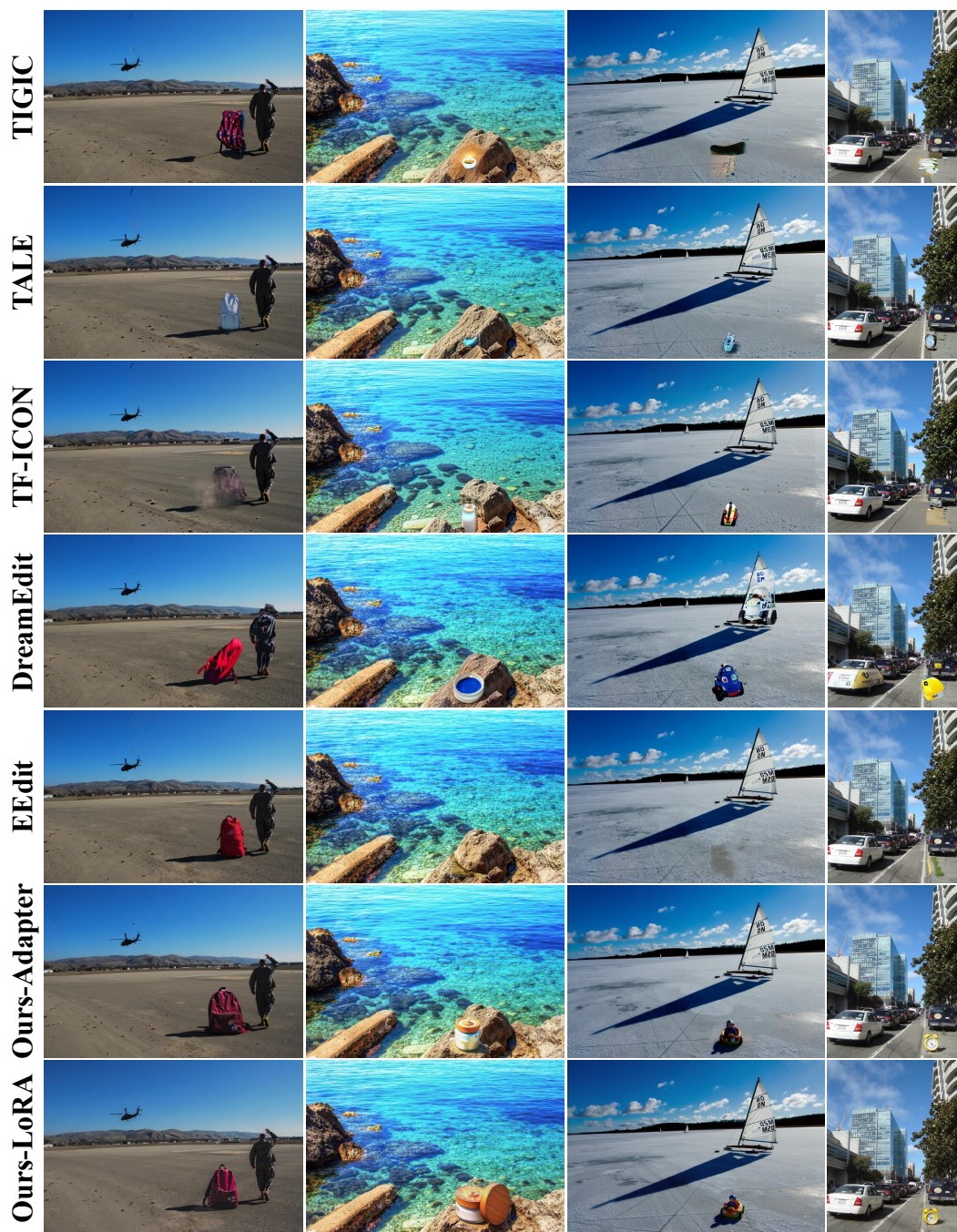

Figure 14: **(Part 2 of 2)** Qualitative comparison of our method against baselines in challenging scenarios.

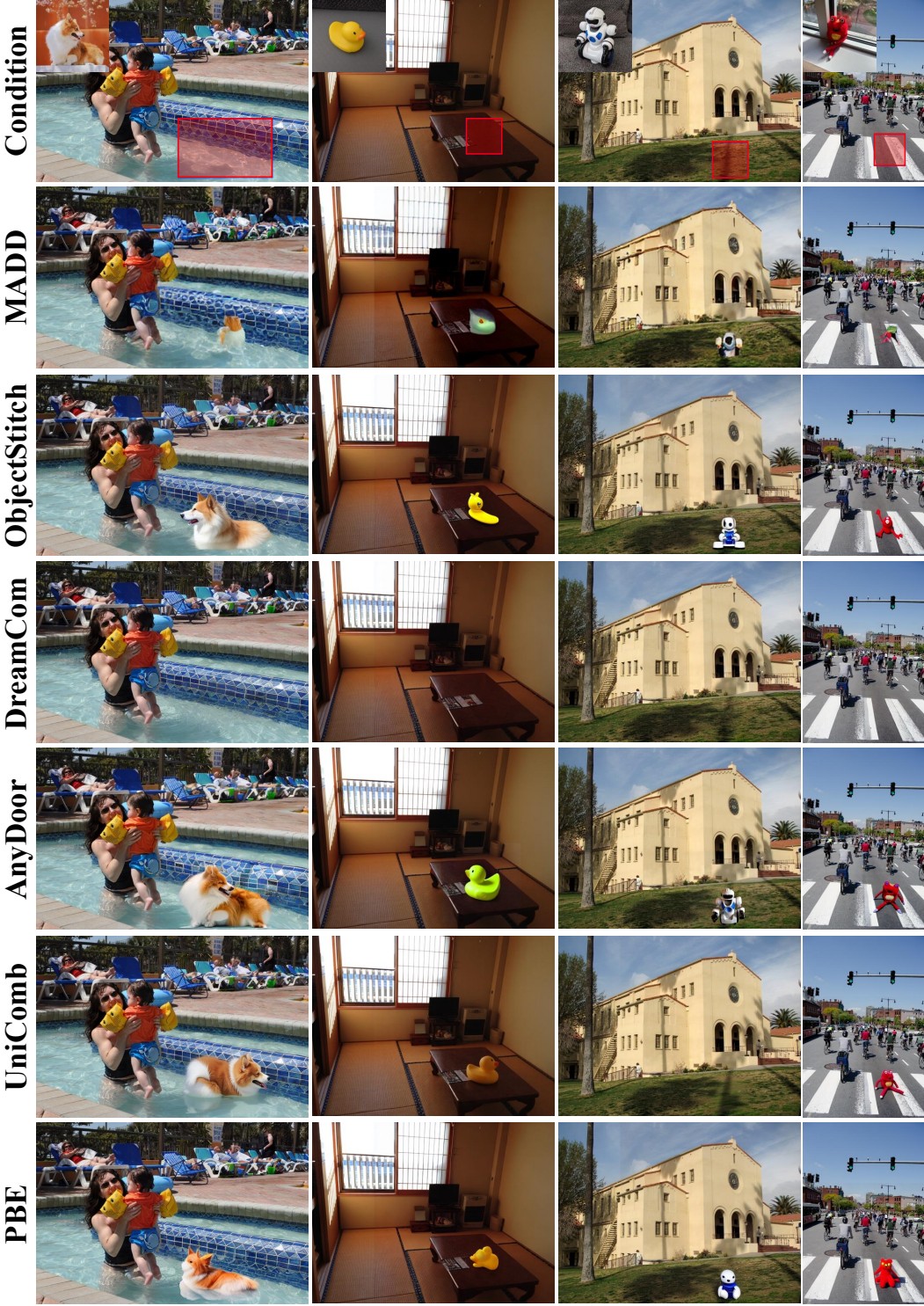

Figure 15: **(Part 1 of 2)** Qualitative comparison of our method against baselines in challenging scenarios.

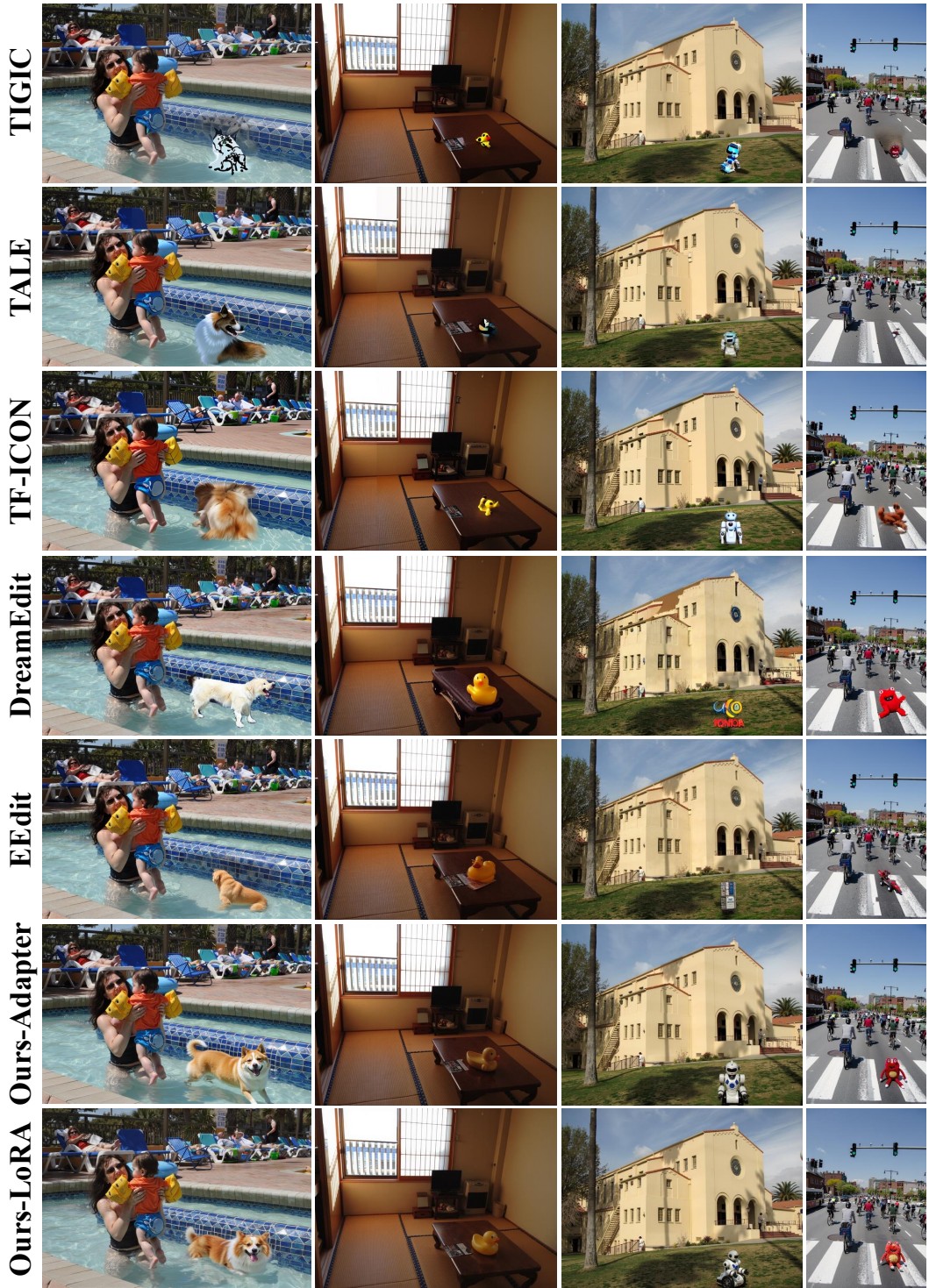

Figure 16: **(Part 2 of 2)** Qualitative comparison of our method against baselines in challenging scenarios.

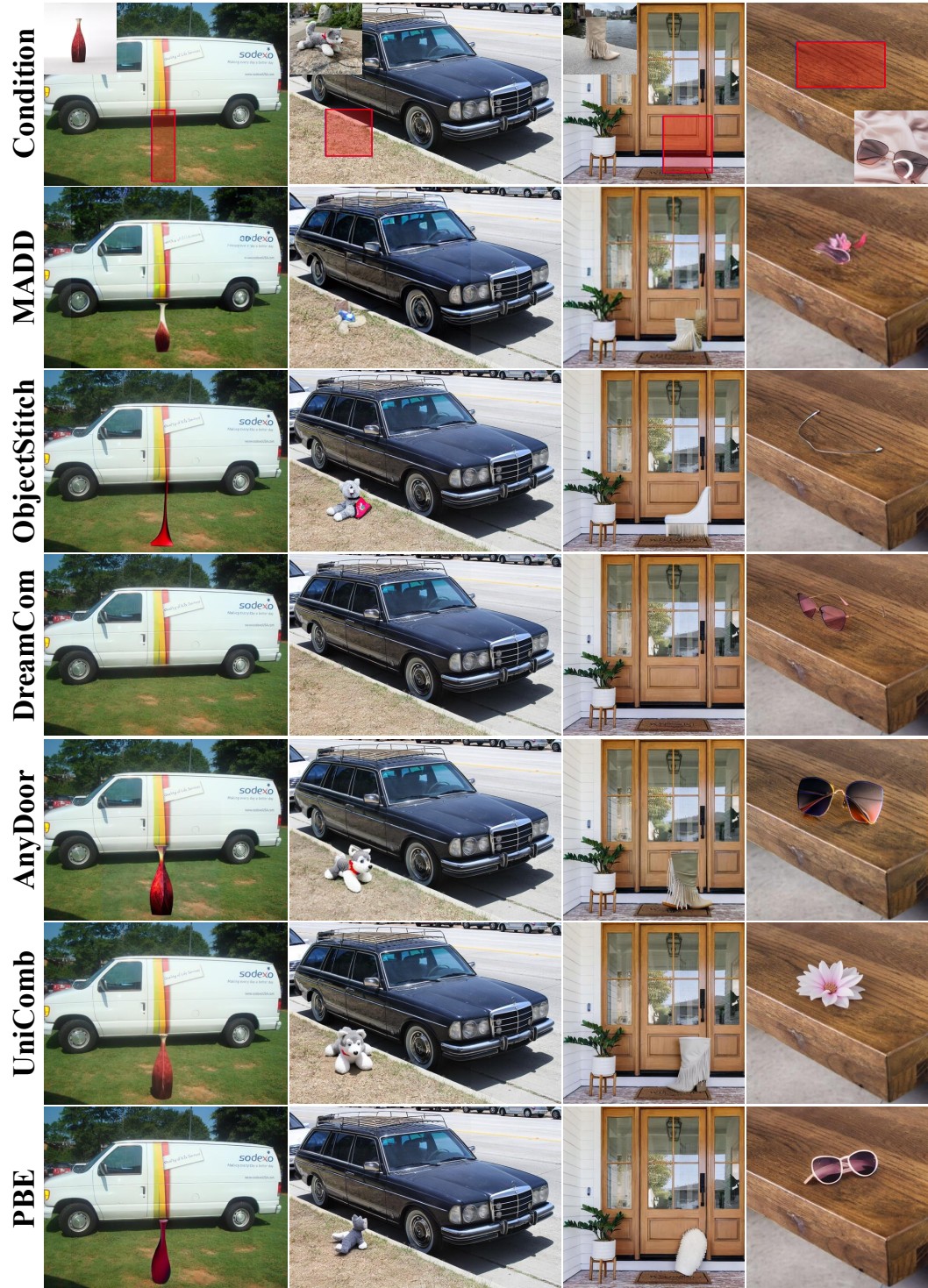

Figure 17: **(Part 1 of 2)** Qualitative comparison of our method against baselines in challenging scenarios.

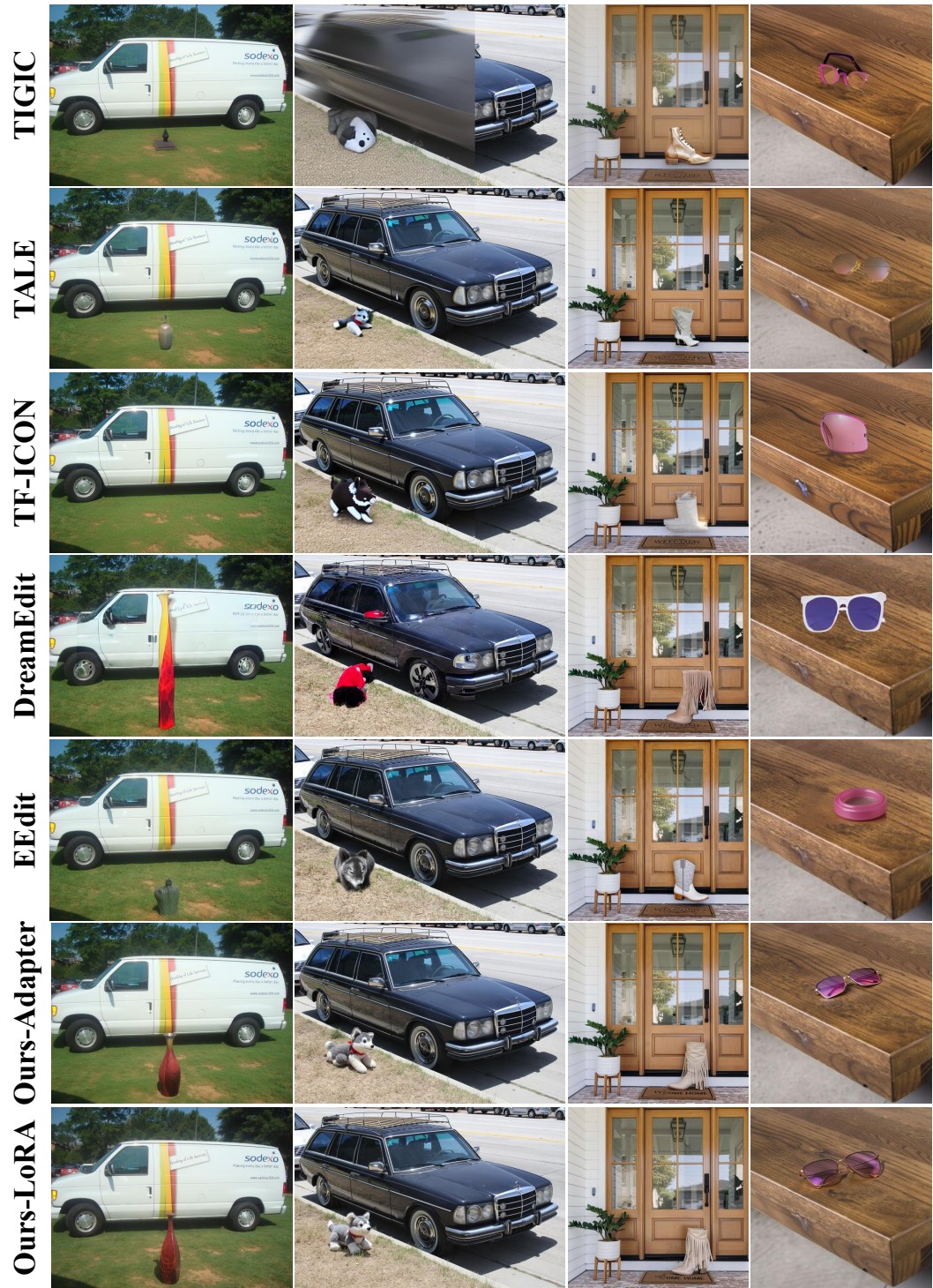

Figure 18: **(Part 2 of 2)** Qualitative comparison of our method against baselines in challenging scenarios.

