# OpenReview forum: "Does FLUX Already Know How to Perform Physically Plausible Image Composition?"
_ICLR.cc/2026/Conference — ICLR 2026 Poster_

### Official Review · Reviewer_Yk7s · 2025-10-23

**Soundness:** 3
**Presentation:** 4
**Contribution:** 3
**Rating:** 8
**Confidence:** 4

**Summary:**

SHINE is a training-free framework designed for seamless, high-fidelity insertion with neutralized errors. It introduces the manifold-steered anchor loss, which leverages pretrained customization adapters to guide latent representations, ensuring accurate subject representation while maintaining the integrity of the background. Additionally, SHINE incorporates degradation-suppression guidance and adaptive background blending to prevent low-quality results and visible seams, enhancing the overall output quality.
In response to the lack of rigorous benchmarks in this domain, the authors propose ComplexCompo, a benchmark suite that tests the framework under challenging conditions.

**Strengths:**

1. The article presents a clear expression of its ideas, with a thorough visualization analysis of the motivation and proposed modules.

2. The experiments are comprehensive, and the results appear highly promising.

3. The work makes notable breakthroughs in addressing issues related to lighting realism and resolution rigidity in image synthesis.

**Weaknesses:**

1. As shown in Table 1, the method performs well in Subject Identity Consistency, but there is room for improvement in Background Retention. Could you provide further insights and possible approaches for enhancing this aspect?

2. In the Image Quality evaluation, the models used for assessment are somewhat limited. Could you consider incorporating more diverse and widely-adopted image quality evaluation models, such as UnifiedReward [1] and HPSv3 [2], to provide a more comprehensive analysis?

[1] Unified Reward Model for Multimodal Understanding and Generation. 2025

[2] HPSv3: Towards Wide-Spectrum Human Preference Score. 2025

**Questions:**

See Weakness.

---

> ### Author Response · Authors · 2025-11-20
> **Response to Reviewer Yk7s (1/2)**
>
> Dear Reviewer Yk7s,
>
> Thank you very much for taking the time to review our manuscript and for offering many valuable comments that helped us improve the work.
>
> We sincerely appreciate your recognition of the clarity of our presentation, the promising results, and our contributions toward addressing challenges related to lighting realism and resolution rigidity in this field.
>
> We will address your concerns point by point.
>
> ---
>
> **[W1] As shown in Table 1, the method performs well in Subject Identity Consistency, but there is room for improvement in Background Retention. Could you provide further insights and possible approaches for enhancing this aspect?**
>
> Thank you very much for this thoughtful question. Below, we provide our insights on the underlying factors, followed by two possible directions for improvement.
>
> 1. **Our insights**
>
>     - For most training-free image composition methods, background blending, $\boldsymbol{z}\_{t}^\prime = \boldsymbol{M} \odot \boldsymbol{z}\_{t} + (1 - \boldsymbol{M}) \odot \boldsymbol{z}^\text{bg}\_{t}$,
>     is applied at every diffusion step except the final one (applying blending at the last step typically causes very noticeable seams along the mask boundary). Thus, the background at the final step is determined by (i) the model’s denoising accuracy at the last step, and (ii) the latent provided to the model at that step.
>
>     - Our method employs an adapter-augmented model. While the adapter does not inherently degrade denoising accuracy, it may slightly affect the denoising trajectory, which can reduce the similarity between the output background and the original one. The use of DSG sampling can also introduce subtle variations in the background.
>
>     - These effects are not consistent across different resolutions. Notably, on the more diverse CompleCompo Bench, containing images with a wide range of resolutions, our method achieves the best and second-best SSIM scores (Table 1).
>
> 2. **Potential approaches for further enhancing background retention**
>
>     At this scale of LPIPS and SSIM, the differences in the background are almost imperceptible to the human eye. However, in scenarios that require very high precision, we can consider two possible directions: (1) mask-aware DSG and (2) perceptual guidance.
>
>     *(i) Mask-aware DSG*
>
>     A straightforward extension is to assign a larger DSG weight $\eta$ to the foreground region and a smaller one to the background region, so that the model’s predictions in the background area are less influenced. Formally, Equation (3) in the manuscript can be adapted as:
>
>     $$ \boldsymbol{v}^{\mathrm{dsg}}\_{t} = \boldsymbol{v}\_{\boldsymbol{\theta}+\Delta\boldsymbol{\theta}}(\boldsymbol{z}\_{t}, t, \boldsymbol{c}, \boldsymbol{z}^\text{subj}) + \boldsymbol{M}\_{\eta} \odot \bigl(
>     \boldsymbol{v}\_{\boldsymbol{\theta}+\Delta\boldsymbol{\theta}}(\boldsymbol{z}\_{t}, t, \boldsymbol{c}, \boldsymbol{z}^\text{subj}) - \boldsymbol{v}^{\mathrm{neg}}\_{\boldsymbol{\theta}+\Delta\boldsymbol{\theta}}(\boldsymbol{z}\_{t}, t, \boldsymbol{c}, \boldsymbol{z}^\text{subj}) \bigr), $$
>
>     where $\boldsymbol{M}_{\eta}$ is a spatial weighting matrix assigning different DSG strengths to different regions.
>
>
>     *(ii) Perceptual guidance*
>
>     Another possible enhancement is to compute a perceptual difference between the predicted background (from an intermediate latent) and the target background:
>     $$
>     L_{\text{bg}} = \left\\| \phi\left((1-\boldsymbol{M}^\text{user})\odot \hat{\boldsymbol{x}}_0\right) \-\
>                     \phi\left((1-\boldsymbol{M}^\text{user})\odot \boldsymbol{x}^{\text{bg}}\right) \right\\|_2,
>     $$
>     where $\phi$ denotes a perceptual feature encoder (e.g., LPIPS or intermediate layers of DINO/CLIP/VGG).
>
>
>     Here, $L_{\text{bg}}$ can be treated as an energy function whose gradient is used to guide the diffusion prediction:
>     $$
>     \tilde{\boldsymbol{v}} = \boldsymbol{v}\_{\boldsymbol{\theta}}(\boldsymbol{z}\_{t}, t, \boldsymbol{c}) + \beta ~\nabla\_{\boldsymbol{z}\_{t}} L\_{\text{bg}}.
>     $$
>
> We sincerely appreciate your thoughtful comments. If you have any additional comments or further suggestions, we would be more than happy to discuss them.

---

> ### Author Response · Authors · 2025-11-20
> **Response to Reviewer Yk7s (2/2)**
>
> **[W2] In the Image Quality evaluation, the models used for assessment are somewhat limited. Could you consider incorporating more diverse and widely-adopted image quality evaluation models, such as UnifiedReward and HPSv3, to provide a more comprehensive analysis?**
>
> Thank you for your suggestion. Your recommendation significantly strengthened our work and made our evaluation more comprehensive.
>
> Following your advice, we have now included three variants of UnifiedReward (UnifiedReward-2.0-qwen3vl-8b, UnifiedReward-Edit-qwen3vl-8b, and UnifiedReward-Think-qwen-7b), along with HPSv3, into our Image Quality evaluation pipeline.
>
> The quantitative results assessed by these models are reported in the tables below. **Moreover, we provide a more comprehensive version of this table, which includes all fine-grained sub-scores, in Table 6 in the revised manuscript.** We have also added a note in the *Metrics* section (Line 410-411) to guide readers to this detailed table. Please feel free to share any additional suggestions or thoughts. We’d be delighted to discuss them with you.
>
> - **Results on DreamEditBench**
>
>     |Method|HPSv3|UnifiedReward-2.0-qwen3vl-8b|UnifiedReward-Edit-qwen3vl-8b|UnifiedReward-Think-qwen-7b|
>     |-|-|:-:|:-:|:-:|
>     |MADD|1.2443|15.4740|13.8148|17.6627|
>     |ObjectStitch|7.4529|19.9499|19.1886|20.0838|
>     |DreamCom|5.9324|20.3132|19.9296|19.3399|
>     |Anydoor|8.4867|20.9852|19.0989|20.8230|
>     |Unicombine|8.8415|21.8307|21.7080|20.5224|
>     |PBE|8.3789|21.3636|20.2137|21.3698|
>     |TIGIC|5.2676|17.8784|17.1000|19.1755|
>     |TALE|6.3773|19.3780|18.0784|20.2555|
>     |TF-ICON|7.2643|19.9382|18.2716|20.0168|
>     |DreamEdit|6.025|19.0659|15.7636|19.5836|
>     |EEdit|6.6689|19.8156|19.5603|20.6720|
>     |Ours-Adapter|**8.8861**|21.9523|**22.0182**|**21.7041**|
>     |Ours-LoRA|8.8688|**21.9727**|21.9421|21.3778|
>
> - **Results on ComplexCompo**
>
>     |Method|HPSv3|UnifiedReward-2.0-qwen3vl-8b|UnifiedReward-Edit-qwen3vl-8b|UnifiedReward-Think-qwen-7b|
>     |-|-|:-:|:-:|:-:|
>     |MADD|5.9673|15.4258|13.0567|17.2524|
>     |ObjectStitch|8.8389|18.6983|18.8283|18.8394|
>     |DreamCom|7.9884|16.1726|18.6143|19.0509|
>     |Anydoor|8.9760|18.4908|18.3550|19.8876|
>     |Unicombine|8.8999|18.1770|19.8529|19.6449|
>     |PBE|8.5923|18.7007|18.9507|19.6170|
>     |TIGIC|7.6630|16.9677|16.4301|18.2956|
>     |TALE|8.7351|17.9581|18.7567|18.7955|
>     |TF-ICON|9.3258|18.2030|18.2366|19.2380|
>     |DreamEdit|8.0434|16.2770|15.8934|18.9805|
>     |EEdit|8.7835|17.7241|19.7348|20.2367|
>     |Ours-Adapter|9.6485|**20.3326**|20.7349|20.9647|
>     |Ours-LoRA|**9.8418**|20.2492|**21.0326**|**21.1212**|

---

> > ### Comment · Reviewer_Yk7s · 2025-11-21
> >
> > For W1: Thank you for the detailed analysis. I look forward to seeing further improvements and refinements of your method.
> >
> > For W2: The image quality evaluation results are convincing. I strongly recommend including these results in the final version of the paper.
> >
> > A minor additional comment: there may be a typographical issue in the model name `UnifiedReward-Think-qwen3-7b`; as far as I know, the Qwen3-VL models do not include a 7B variant, so please double-check this.
> >
> > Overall, I think the authors have provided a strong rebuttal and have adequately addressed my concerns. I recommend accepting this work.

---

> > > ### Author Response · Authors · 2025-11-21
> > > **Response to Reviewer Yk7s**
> > >
> > > Thank you so much for your encouraging follow-up. We truly appreciate your constructive feedback.
> > >
> > > Regarding W2, we have now incorporated the full image quality evaluation results into Table 6. In addition, we have further expanded Table 1 by adding two new columns reporting the results of HPSv3 and UnifiedReward-Edit-qwen3vl-8b. If you have any suggestions for a better way to present or organize these numbers, we would be very grateful to follow them.
> > >
> > > Thank you as well for pointing out the typographical issue. The correct model name we used is UnifiedReward-Think-**qwen**-7b [1] rather than **qwen3**-7b, and we have updated this accordingly.
> > >
> > > We sincerely appreciate your positive assessment and your recommendation for acceptance. Thank you again for your time, support, and helpful comments.
> > >
> > > [1] https://huggingface.co/CodeGoat24/UnifiedReward-Think-qwen-7b

---

### Official Review · Reviewer_5xsu · 2025-10-31

**Soundness:** 4
**Presentation:** 3
**Contribution:** 3
**Rating:** 8
**Confidence:** 5

**Summary:**

The paper presents SHINE, a training-free strategy for image composition using FLUX. Since previous inversion-based and attention manipulation–based approaches struggle to achieve robust image composition, the authors introduce Non-Inversion Latent Preparation, Manifold-Steered Anchor (MSA) Loss, Degradation-Suppression Guidance (DSG), and Adaptive Background Blending (ABB). FLUX.1-dev is a CFG-distilled model, which causes high inversion errors. The subject image is captioned by a VLM, and an inpainting model creates an initial image based on the caption. Building on this, the authors perform a one-step forward diffusion to obtain and manipulate the latent. The Manifold-Steered Anchor (MSA) Loss utilizes the velocity predicted by the original noisy latent as anchor noise, along with the adapter-guided velocity prediction using the subject image’s latent. Degradation-Suppression Guidance (DSG) is inspired by negative prompting and constructs a negative velocity prediction by blurring the attention component $Q_{img}$ of FLUX. Adaptive Background Blending (ABB) selects between the attention mask and a user-provided mask depending on the timestep. Experiments demonstrate the effectiveness of the proposed framework for image composition, both quantitatively and qualitatively. The authors also propose ComplexCompo, a benchmark for challenging composition scenarios.

**Strengths:**

1. The design of the Manifold-Steered Anchor Loss, which takes into account the flow-matching process between FLUX.1-dev and the adapter model (InstantCharacter), is well-motivated and technically sound.
2. The in-depth analysis of the attention blocks in FLUX.1-dev (Sec. C & D) and the derivation of the subsequent improvement, Degradation-Suppression Guidance, are particularly interesting.
3. The Adaptive Background Blending method is simple yet powerful, and it appears broadly applicable to various image editing tasks.
4. The paper offers numerous insights drawn from a comprehensive set of dense experiments.
5. The proposed methods are model- and adapter-agnostic, making them broadly applicable across different architectures and integration settings.

**Weaknesses:**

Overall, the paper is still a strong and well-executed, however, it could be further improved if the authors provide additional details and clarifications on certain points regarding the following aspects:

1. In L159-161, the term “inpainted background image” appears somewhat inappropriate in this context. Phrases such as “scene image” or “image to which the subject is attached” would be more accurate and better convey the intended meaning.

2. In L186-187, the authors state that they abandon the inversion process and instead apply one-step forward diffusion. I interpret this as implying that “the effect of inversion can be approximated by directly adding noise corresponding to a one specific timestep”. If this interpretation is correct, a theoretical justification should be provided. Additionally, the authors should clarify how this approach differs from the inversion-skipping technique used in EEdit [1].

3. While numerical improvements are observed in Config A and C of the ablation study, it remains unclear whether DSG truly acts as a negative guidance vector when blurring the attention component $Q_{img}$ derived from the text embedding $c$ extracted by the VLM. Although FLUX is a CFG-distilled model where non-sensical or explicit negative prompts are often ineffective (as shown in Fig. 4(a)), this behavior may differ in other MM-DiT–based models such as SD3.5. It would strengthen the paper to include an analysis comparing simple negative prompting and Degradation-Suppression Guidance in SD3.5 to better quantify their differences.

4. Please specify which inpainting model (e.g., Brushedit, BrushNet, PowerPaint, Flux.1 fill-dev) and which VLM (e.g., BLIP-3, InternVL, LLaVA-1.5) were used in the proposed frame work and evaluation.

5. In Alg. 1, the origin of $z^{bg}$ is unclear. Is it defined as $z^{bg} = (1 - M^{user}) \odot z^{init}$ and $z^{subject} =  M^{user} \odot z^{init}$? Please clarify this in the text or Alg. 1.

6. In Tab. 1 & 3, it would be beneficial to include results for Config A (the input image generated by the inpainting model). This addition would help readers better understand the performance gap and the impact of subsequent modules at a glance.

[1] EEdit : Rethinking the Spatial and Temporal Redundancy for Efficient Image Editing, ICCV 2025

**Questions:**

1. In L260-262, how are the attention components systematically perturbed? Which 2D Gaussian filter $G$ (kernel size) is used?
2. How much time and GPU memory are required to generate a composited image using FLUX.1-dev + Adapter and + LoRA? Also, for the other baselines?

**Details Of Ethics Concerns:**

No concern.

---

> ### Author Response · Authors · 2025-11-20
> **Response to Reviewer 5xsu (1/3)**
>
> Dear Reviewer 5xsu,
>
> Thank you very much for taking the time to review our manuscript and for offering many valuable suggestions that helped us improve the work.
>
> We sincerely appreciate your recognition that our MSA loss is well-motivated and technically sound, DSG is interesting, ABB is simple yet powerful, and that all these components are both model- and adapter-agnostic.
>
> We will address your concerns point by point.
>
> ---
> **[W1] In L159-161, the term “inpainted background image” appears somewhat inappropriate in this context. Phrases such as “scene image” or “image to which the subject is attached” would be more accurate and better convey the intended meaning.**
>
> Thank you for your suggestion. We agree that the term “inpainted background image” may not be the most appropriate in this context. Accordingly, we have revised it to “image to which the subject is attached” (Line 161 of updated version), as you recommended, to better convey the intended meaning and improve clarity.
>
> ---
> **[W2] In L186-187, the authors state that they abandon the inversion process and instead apply one-step forward diffusion. I interpret this as implying that “the effect of inversion can be approximated by directly adding noise corresponding to a one specific timestep”. If this interpretation is correct, a theoretical justification should be provided. Additionally, the authors should clarify how this approach differs from the inversion-skipping technique used in EEdit.**
>
> Thank you for this thoughtful comment. Below, we address: (1) why we use one-step forward diffusion instead of inversion (even though they are not theoretically equivalent), and (2) how our approach differs from EEdit’s inversion-skipping strategy.
>
> 1. **Why we replace inversion with one-step forward diffusion (even though they are not theoretically equivalent)?**
>
>     - We use one-step forward diffusion as a practical substitute for inversion, not because the two operations are theoretically equivalent. Our approach deliberately avoids relying on the initial latent to preserve the object’s identity. Instead, the MSA loss extracts the object information from the adapter/LoRA and injects it into the latent during editing.
>     - This design choice is motivated by a practical observation: many modern models are distilled for speed, making accurate inversion difficult to achieve in practice. **When inversion cannot reliably encode object identity, its benefit becomes limited. In such cases, a noisy latent that still retains enough background structure is sufficient as a starting point.** For this reason, we adopt one-step forward diffusion as a pragmatic replacement for inversion, rather than a theoretically equivalent alternative. It offers two advantages: (1) it avoids the accuracy limitations of inversion on distilled models, making it more broadly applicable, and (2) it is faster than performing an inversion.
>
> 2. **Difference from the inversion-skipping in EEdit.**
>
>     - The key difference is that our method is **non-inversion-based and avoids any model prediction**, while EEdit is inherently **inversion-based and dependent on model predictions**.
>
>     - In EEdit, the initial latent is still obtained via an inversion procedure that involves running the model to predict the latent. The proposed “inversion skipping” is a very elegant and effective technique to greatly accelerate this inversion process by skipping several steps.
>
>     - In contrast, our one-step forward diffusion does **not** perform inversion at all. The initial latent is produced by directly sampling noise and adding it to the clean latent at a chosen timestep, without any model prediction. This makes our initialization computationally lighter.
>
>     - Because FLUX-Dev is a CFG-distilled model, its inversions are relatively imprecise, which we believe contributes to the weaker subject-identity preservation observed in EEdit. Our method is therefore designed to avoid reliance on accurate inversion in such distilled settings.
>
> We have incorporated these clarifications into Appendix K of the revised version. We sincerely appreciate your comments, which have helped us improve the clarity and presentation of our work.

---

> ### Author Response · Authors · 2025-11-20
> **Response to Reviewer 5xsu (2/3)**
>
> **[W3] While numerical improvements are observed in Config A and C of the ablation study, it remains unclear whether DSG truly acts as a negative guidance vector when blurring the attention component $Q_{img}$ derived from the text embedding $c$ extracted by the VLM. Although FLUX is a CFG-distilled model where non-sensical or explicit negative prompts are often ineffective (as shown in Fig. 4(a)), this behavior may differ in other MM-DiT–based models such as SD3.5. It would strengthen the paper to include an analysis comparing simple negative prompting and Degradation-Suppression Guidance in SD3.5 to better quantify their differences.**
>
> Thank you for raising this important point. You are correct that in SD3.5, negative prompts can indeed produce visibly degraded or low-quality images.
>
> Following your suggestion, we conduct experiments on SD3.5 by replacing our DSG with standard negative prompting (NP). Specifically, we replaced the negative velocity $\boldsymbol{v}^{\text{neg}}_{\boldsymbol{\theta}+\Delta\boldsymbol{\theta}}$ in Eqn. 3 with the velocity obtained using the following four commonly used negative prompts:
> - NP 1: distorted, deformed, glitch, artifacts
> - NP 2: undefined shapes, bad anatomy, unnatural pose
> - NP 3: low quality, worst quality, low resolution, blurry, out of focus
> - NP 4: AI artifacts, melted objects, strange textures
>
> The quantitative results on the DreamEditBench are reported below:
> |Method|CLIP ↑|DINO ↑|IRF ↑|DreamSim ↓|LPIPS ↓|SSIM ↑|IR ↓|VR ↑|
> |-|-|-|-|-|-|-|-|-|
> |Ours-SD3.5-Adapter (w/ NP 1)|0.7934|0.7311|0.7608|0.3918|0.0331|0.8812|0.5837|3.5714|
> |Ours-SD3.5-Adapter (w/ NP 2)|0.8029|0.7388|0.7679|0.3781|0.0259|0.8908|0.5726|3.6102|
> |Ours-SD3.5-Adapter (w/ NP 3)|0.7958|0.7329|0.7623|0.3892|0.0315|0.8839|0.5814|3.5796|
> |Ours-SD3.5-Adapter (w/ NP 4)|0.7887|0.7281|0.7574|0.3976|0.0368|0.8765|0.5882|3.5548|
> |Ours-SD3.5-Adapter (w/ DSG)|**0.8054**|**0.7407**|**0.7699**|**0.3745**|**0.0234**|**0.8937**|**0.5701**|**3.6187**|
>
> These results reveal several key insights:
> 1. Performance varies across different negative prompts, confirming that the effectiveness of heuristic negative prompts is sensitive to their wording.
> 2. Negative prompts generate degradation directions that are decoupled from the specific input image, whereas DSG constructs an image-specific low-quality direction. This makes DSG more adaptive, stable, and effective across diverse inputs without requiring manual prompt engineering.
>
> We have incorporated these experiments into Appendix M. If you have further suggestions, we would be glad to continue the discussion. Your feedback is genuinely helpful in strengthening our work.
>
> ---
> **[W4] Please specify which inpainting model (e.g., Brushedit, BrushNet, PowerPaint, Flux.1 fill-dev) and which VLM (e.g., BLIP-3, InternVL, LLaVA-1.5) were used in the proposed frame work and evaluation.**
>
> Thank you for pointing this out and for helping us improve the manuscript.
>
> In our experiments, we use Flux.1 fill-dev as the inpainting model and BLIP-3 as the VLM. We apologize for this earlier omission and have revised the *Implementation Details* section (Line 420) to clarify these details.
>
> ---
> **[W5] In Alg. 1, the origin of $z^{bg}$ is unclear. Is it defined as $z^{bg} = (1 - M^{user}) \odot z^{init}$ and $z^{subject} = M^{user} \odot z^{init}$? Please clarify this in the text or Alg. 1.**
>
> Thank you for the comment. There is indeed a potential misunderstanding regarding how the latents are constructed during preprocessing, and we now clarify this in the Algorithm 1.
>
> The user provides three inputs:
> 1. Background image $\boldsymbol{x}^{\text{bg}}$
> 2. Subject image $\boldsymbol{x}^{\text{subj}}$
> 3. User mask $\boldsymbol{M}^{\text{user}}$
>
> We encode the background and subject images using the VAE encoder $ E(\cdot) $ to obtain
> $$
> \boldsymbol{z}^{\text{bg}} = E(\boldsymbol{x}^{\text{bg}}), \quad
> \boldsymbol{z}^{\text{subj}} = E(\boldsymbol{x}^{\text{subj}}).
> $$
> These two latents are directly obtained from their respective images. They are **not** computed by masking the latent $\boldsymbol{z}^{\text{init}}$.
>
> The latent $\boldsymbol{z}^{\text{init}}$ comes from a separate inpainting step. Specifically, we first use a VLM to caption the subject image and obtain a text prompt. We then inpaint the masked region of the background image:
> $$
> \boldsymbol{x}^{\text{init}}
> = \text{Inpainting}\bigl(\boldsymbol{x}^{\text{bg}}, \boldsymbol{M}^{\text{user}}, \text{prompt}=\text{VLM}(\boldsymbol{x}^{\text{subj}})\bigr),
> $$
> and encode the resulting image:
> $$
> \boldsymbol{z}^{\text{init}} = E(\boldsymbol{x}^{\text{init}}).
> $$
> We have revised the input field of Algorithm 1 to explicitly clarify the definitions of $\boldsymbol{z}^{\text{bg}}$, $\boldsymbol{z}^{\text{subj}}$, and $\boldsymbol{z}^{\text{init}}$. We sincerely appreciate your comment, which helped us improve the clarity of the manuscript.

---

> ### Author Response · Authors · 2025-11-20
> **Response to Reviewer 5xsu (3/3)**
>
> **[W6] In Tab. 1 & 3, it would be beneficial to include results for Config A (the input image generated by the inpainting model). This addition would help readers better understand the performance gap and the impact of subsequent modules at a glance.**
>
> Thank you for the great suggestion. We agree that including the results for Config A makes it easier for readers to understand the performance gap and the impact of the subsequent modules at a glance. We have therefore added the Config A results to Tables 1 and 3 in the revised manuscript.
>
> ---
> **[Q1] In L260-262, how are the attention components systematically perturbed? Which 2D Gaussian filter $G$ (kernel size) is used?**
>
> - Thank you for the question. In our experiments, we apply the same 2D Gaussian blur procedure independently to $\boldsymbol{Q}\_\text{txt}, \boldsymbol{K}\_\text{txt}, \boldsymbol{V}\_\text{txt}, \boldsymbol{Q}\_\text{img}, \boldsymbol{K}\_\text{img}, \boldsymbol{V}\_\text{img}$. For each run, we choose a single component, reshape it into a 2D spatial feature map, apply the blur, reshape it back into sequence form, and then compute attention using the perturbed tensor while keeping all other components unchanged.
>
> - Below we provide pseudocode illustrating how we blur $\boldsymbol{Q}_\text{img}$. For the 2D Gaussian filter, we use the implementation from the *kornia* library. Following common engineering practice, we set the kernel radius $r = 3\sigma$, since three sigmas cover roughly 99.7% of the Gaussian’s energy. Thus, the kernel size is chosen as the nearest odd number around $2r = 6\sigma$. In our experiments, $\sigma = 10$ (Table 5).
>
> ```
> # --- Step 1: Reshape query into spatial form ---
> Query = Query.reshape(B, H, W, D).permute(0, 3, 1, 2)
>
> # --- Step 2: Apply 2D Gaussian blur to spatial feature maps ---
> sigma = 10
> kernel_size = 6 * sigma
> kernel_size = kernel_size - (kernel_size % 2) + 1  # ensure an odd kernel size
> kernel_size = (kernel_size, kernel_size)
> sigma = (sigma, sigma)
>
> Query = kornia.filters.gaussian_blur2d(Query, kernel_size, sigma)
>
> # --- Step 3: Reshape blurred query back to sequence form ---
> Query = Query.permute(0, 2, 3, 1).reshape(B, L, D)
>
> # --- Step 4: Compute attention ---
> attn_weights = softmax(Query @ Key.T / sqrt(D))
> output = attn_weights @ Value
> ```
>
> ---
> **[Q2] How much time and GPU memory are required to generate a composited image using FLUX.1-dev + Adapter and + LoRA? Also, for the other baselines?**
>
> Thank you for your question. The wall-clock runtime at a $512\times512$ resolution on an H100 GPU is summarized in the table below. In our implementation, we applied qint8 quantization to all FLUX-based methods to accelerate inference and reduce GPU memory usage. Additionally, Ours-Adapter can be run on a 24-GB GPU by enabling CPU offloading. We have included this information as Table 8 in the revised manuscript. We sincerely appreciate your feedback, which has helped us further improve the completeness of our work.
> |Method|Training-Free|Base|External Model|Time (s)|Peak Memory (MB)|
> |-|:-:|:-:|:-:|:-:|:-:|
> |MADD|No|SD|DINO|45.73|11708|
> |ObjectStitch|No|SD|VIT|6.63|8268|
> |DreamCom|No|SD|LoRA|9.87|3388|
> |Anydoor|No|SD|DINO|8.61|18612|
> |Unicombine|No|FLUX|LoRA|11.98|22711|
> |PBE|No|SD|-|3.52|10842|
> |TIGIC|Yes|SD|-|10.82|21640|
> |TALE|Yes|SD|-|8.03|23524|
> |TF-ICON|Yes|SD|-|24.55|20670|
> |DreamEdit|Yes|SD|LoRA,VIT|99.83|19298|
> |EEdit|Yes|FLUX|-|60.31|26546|
> |Ours-Adapter|Yes|FLUX|Adapter|38.29|32552|
> |Ours-LoRA|Yes|FLUX|LoRA|18.08|23519|

---

> ### Comment · Reviewer_5xsu · 2025-11-25
>
> Thanks for the authors’ response. I have read it as well as the other reviews. The rebuttal has sufficiently addressed my concerns. I only have a bit of additional minor trimming to suggest.
>
> 1. Please double-check the citations (bibtex) to confirm which papers were accepted by specific conferences. Currently, too many papers published in conferences are listed as arXiv. It would be better to specify the exact conference or journal name where the cited paper was published. For examples:
>
> - In L510, replace "BFL" to "Black Forest Labs".
>   - BFL. Flux.1 fill [dev]. https://huggingface.co/black-forest-labs/FLUX.1-Fill-dev, 2024.
>   - Black Forest Labs. Flux.1 [dev]. https://huggingface.co/black-forest-labs/FLUX.1-dev, 2024a.
>   - Black Forest Labs. Flux.1 [schnell]. https://huggingface.co/black-forest-labs/FLUX.1-schnell, 2024b.
> - Unify the reference format between L518 and L525 (including the abbreviation of conference and URL or not).
>   - Tim Brooks, Aleksander Holynski, and Alexei A. Efros. Instructpix2pix: Learning to follow image editing instructions. 2023 IEEE/CVF Conference on Computer Vision and Pattern Recognition (CVPR), pp. 18392–18402, 2022. URL https://api.semanticscholar.org/CorpusID:253581213.
>   - Bor-Chun Chen and Andrew Kae. Toward realistic image compositing with adversarial learning. In Proceedings of the IEEE/CVF Conference on Computer Vision and Pattern Recognition, pp. 8415–8424, 2019.
> - In L530, change the venue from arXiv to the published venue, ICLR.
>   - Junsong Chen, Jincheng Yu, Chongjian Ge, Lewei Yao, Enze Xie, Yue Wu, Zhongdao Wang, James Kwok, Ping Luo, Huchuan Lu, et al. Pixart-α: Fast training of diffusion transformer for photorealistic text-to-image synthesis. arXiv preprint arXiv:2310.00426, 2023.
> - Remove the duplicated references.
>   - L569 and L573: An Image is Worth One Word
>   - L598 and L602: LoRA
>   - L763 and L767: Dreambooth
>   - L800 and L804: ObjectStitch
>   - L878 and L884: xGen-MM (BLIP-3)
>   - L891 and L894: Paint by Example
>   - L903 and L905: IP-Adapter
>
> 2. The context of the newly added Sec. K is a bit hard to understand without the discussion between the authors and me on OpenReview. The sentence at L1510, “The key difference between our methods and EEdits”, feels like it appears rather suddenly. I think it would be better to add a brief explanation to motivate why you introduce this point here.
>
> 3. I believe the details of the 2D Gaussian filter (Response to Reviewer 5xsu (3/3)) are important for reproducibility. Including this information in Sec. C would be beneficial.

---

> ### Author Response · Authors · 2025-11-26
> **Response to Reviewer 5xsu**
>
> Dear Reviewer 5xsu,
>
> Thank you so much for your careful reading and the thoughtful follow-up suggestions. We appreciate the time you devoted to reviewing both our manuscript and rebuttal. Your comments not only helped us address the remaining issues but also guided us to further polish the paper in ways we had overlooked.
>
> 1. Regarding the **citation formats**, your guidelines are important for improving the accuracy and consistency of the bibliography. We have followed your suggestions and carefully reviewed each reference. The corrections made in this revision are summarized below:
>
>     ```
>     a. Replace wording:
>     - Line L510: Replace “BFL” with “Black Forest Labs”.
>
>     b. Remove URLs from the following BibTeX entries:
>     - “InstructPix2Pix: Learning to Follow Image Editing Instructions”
>     - “SAM 2: Segment Anything in Images and Videos”
>     - “Instructedit: Improving Automatic Masks for Diffusion-Based Image Editing with User Instructions”
>     - “Multimodal Representation Alignment for Image Generation: Text–Image Interleaved Control Is Easier Than You Think”
>     - “SmartEdit: Exploring Complex Instruction-Based Image Editing with Multimodal Large Language Models”
>     - “BrushEdit: All-In-One Image Inpainting and Editing”
>     - “X2i: Seamless Integration of Multimodal Understanding into Diffusion Transformer via Attention Distillation”
>
>     c. Update publication venues (arXiv → specified venue):
>     - “PixArt-α: Fast Training of Diffusion Transformer for Photorealistic Text-to-Image Synthesis” → ICLR
>     - “DreamSim: Learning New Dimensions of Human Visual Similarity Using Synthetic Data” → NeurIPS
>     - “An Image is Worth One Word: Personalizing Text-to-Image Generation Using Textual Inversion” → ICLR
>     - “ACE: All-round Creator and Editor Following Instructions via Diffusion Transformer” → ICLR
>     - “LoRA: Low-Rank Adaptation of Large Language Models” → ICLR
>     - “DreamEdit: Subject-Driven Image Editing” → TMLR
>     - “Gligen: Open-Set Grounded Text-to-Image Generation” → CVPR
>     - “Painterly Image Harmonization Using Diffusion Model” → ACM MM
>     - “SDEdit: Guided Image Synthesis and Editing with Stochastic Differential Equations” → ICLR
>     - “DINOv2: Learning Robust Visual Features Without Supervision” → TMLR
>     - “SDXL: Improving Latent Diffusion Models for High-Resolution Image Synthesis” → ICLR
>     - “DreamFusion: Text-to-3D Using 2D Diffusion” → ICLR
>     - “Magic Insert: Style-Aware Drag-and-Drop” → ICCV
>     - “Denoising Diffusion Implicit Models” → ICLR
>     - “Harmonizing Visual and Textual Embeddings for Zero-Shot Text-to-Image Customization” → AAAI
>     - “Thinking Outside the BBox: Unconstrained Generative Object Compositing” → ECCV
>     - “Multitwine: Multi-Object Compositing with Text and Layout Control” → CVPR
>     - “Add-it: Training-Free Object Insertion in Images With Pretrained Diffusion Models” → ICLR
>     - “ObjectMate: A Recurrence Prior for Object Insertion and Subject-Driven Generation” → ICCV
>     - “OmniGen: Unified Image Generation” → CVPR
>     - “Mastering Text-to-Image Diffusion: Recaptioning, Planning, and Generating with Multimodal LLMs” → ICML
>     - “AnyEdit: Mastering Unified High-Quality Image Editing for Any Idea” → CVPR
>     - “X2i: Seamless Integration of Multimodal Understanding into Diffusion Transformer via Attention Distillation” → ICCV
>
>     d. Remove duplicate entries:
>     - “An Image is Worth One Word: Personalizing Text-to-Image Generation Using Textual Inversion”
>     - “BrushEdit: All-in-One Image Inpainting and Editing”
>     - “DreamBooth: Fine-Tuning Text-to-Image Diffusion Models for Subject-Driven Generation”
>     - “ObjectStitch: Object Compositing with Diffusion Model”
>     - “xgen-mm (blip-3): A Family of Open Large Multimodal Models”
>     - “Paint by Example: Exemplar-Based Image Editing with Diffusion Models”
>     - “LoRA: Low-Rank Adaptation of Large Language Models”
>     - “Ip-Adapter: Text Compatible Image Prompt Adapter for Text-to-Image Diffusion Models”
>
>     e. Add missing bibliographic information:
>     - “Magicapture: High-Resolution Multi-Concept Portrait Customization”: Add volume, issue number, and page range.
>     ```
>
> 2. Regarding **Sec. K**, we have added an explanation in the first paragraph to clarify the motivation for discussing the design choice between inversion and one-step forward diffusion, thereby enhancing the reader’s understanding. We have also refined the transition to more smoothly introduce the differences between EEdit and our approach.
>
> 3. For **Sec. C**, we have now included the details of the 2D Gaussian filter, and we have incorporated the corresponding pseudocode into **Algorithm 2** to improve the completeness of that section.
>
> Once again, thank you for your dedication and for helping us improve the manuscript. Your thoughtful feedback has played an important role in shaping a more polished version. If you have any further suggestions, please feel free to let us know.
>
> With sincere appreciation,
>
> The Authors

---

> ### Comment · Reviewer_5xsu · 2025-11-28
>
> Thank you for revising the paper as suggested.
>
> I think it would be better to remove L1499, “Motivated by the thoughtful feedback from Reviewer 5xsu (see the detailed discussion on OpenReview),” since Reviewer 5xsu is not part of the paper itself. Even if remove L1499, the current Sec. K reads smoothly. The readers will be able to grasp sufficient context from the paper alone.
>
> Overall, the contributions of Manifold-Steered Anchor (MSA) Loss, Degradation-Suppression Guidance (DSG), and Adaptive Background Blending (ABB) are substantial. Although the proposed techniques are developed for the image composition task, I believe each of them can be utilized in other tasks:
>
> - MSA Loss → model-agnostic bridging between pretrained diffusion models and adapter models (image customization)
> - DSG → a replacement for negative CFG (image generation)
> - ABB → applicable to all mask-based inpainting
>
> I appreciate the additional experiments conducted for the rebuttal, and for the reasons mentioned above, I will raise my rating from 8 to 10.

---

> ### Comment · Reviewer_5xsu · 2025-11-28
>
> Since current OpenReview system is unstable, I cannot edit my rating. I will update the rating as soon as it stabilizes. Or if it is no longer possible to modify, I hope AC will reflect this final score.

---

> > ### Author Response · Authors · 2025-11-28
> >
> > Dear Reviewer 5xsu,
> >
> > Thank you so much for your positive comments and for taking the time to assess our revision. We truly appreciate your thoughtful guidance and encouragement.
> >
> > As you suggested, we have removed L1499 in the latest revision.
> >
> > We are delighted to hear that the additional experiments were helpful, and we sincerely appreciate your willingness to raise your rating. No matter how the system handles the final update, your support means a lot to us.
> >
> > Wishing you all the best,
> >
> > The Authors

---

### Official Review · Reviewer_6v7A · 2025-11-01

**Soundness:** 3
**Presentation:** 3
**Contribution:** 3
**Rating:** 6
**Confidence:** 2

**Summary:**

This paper addresses the challenge of physically plausible image composition, particularly the failure of existing models to handle complex lighting, shadows, and reflections. It propose SHINE, a "training-free" framework that is effectively a test-time optimization process.
The framework consists of three main components:
1) Manifold-Steered Anchor (MSA) Loss, a guidance loss that uses a pretrained adapter (e.g., IP-Adapter) and the base model's predictions as an anchor to steer the latent variable;
2) Degradation-Suppression Guidance (DSG), a form of negative guidance achieved by blurring specific query features ($Q_{img}$) inside the DiT to avoid artifacts;
3) Adaptive Background Blending (ABB), a technique that uses cross-attention maps in early sampling steps to create smoother blends.

Additionally, the authors contribute a new benchmark for evaluating composition in complex lighting conditions.

**Strengths:**

1.  Good Qualitative Results: The qualitative results presented (e.g., Fig. 1, 6) are impressive, demonstrating a high degree of physical realism in challenging scenarios.
2.  Benchmark Contribution: The newly proposed ComplexCompo dataset is a solid contribution, offering a valuable tool for future research.

**Weaknesses:**

1.  Impractical Computational Cost: The "training-free" claim is misleading, as the method is a test-time optimization with *extreme computational costs*. Algorithm 1 and Table 5 indicate that it requires $k=10$ optimization iterations *per* denoising step $t$. For a 20-step process, this amounts to ~*200* forward and backward passes. On a 12B parameter model like FLUX, this cost is computationally prohibitive and makes any comparison against single-forward-pass baselines impractical.
2.  Brittle Pre-processing Pipeline: To avoid image inversion, the paper introduces a (Sec 3.1) pre-processing pipeline that depends on *two* external models: a VLM for captioning and an inpainting model for creating $x^{init}$.  If the VLM caption is wrong (e.g., wrong color), the entire composition fails (as indicated in Sec 5-Limitation). It seems that this merely trades one failure point (inversion) for two new, uncontrolled failure points (VLM failure, inpainting failure)?
3.  Limited Novelty and Model-Specific:
    * MSA Loss: Re-naming a known technique (SDS) applied between an adapter and a base model does not constitute a significant novel contribution.
    * DSG: This is an empirical, model-specific hack (via brute-force experimentation) for the FLUX architecture, it seems to lack a theoretical justification and guarantee of generalizability.

**Questions:**

1.  Compute Cost Quantification: Could you provide a fair, wall-clock time comparison on identical hardware, e.g., how much time does SHINE (with $k=10$) take to generate one 1024x1024 image, versus the time required by baselines like AnyDoor or UniCombine?
2.  DSG Generalizability: You claim in Appendix E that DSG applies to SDXL, SD3.5, and PixArt. Was the "blur $Q_{img}$" heuristic applicable *out-of-the-box* to all these architectures, or did each one require a *new* round of brute-force experimentation to find a *different* hack (e.g., "blur $K_{txt}$")?
3.  Pre-processing Failure Rate: I am curious about the actual failure rate of your [VLM + Inpainting pre-processing pipeline], when running the 300-sample ComplexCompo benchmark.

---

> ### Author Response · Authors · 2025-11-20
> **Response to Reviewer 6v7A (1/3)**
>
> Dear Reviewer 6v7A,
>
> Thank you very much for taking the time to review our manuscript and for offering many valuable comments that helped us improve the work.
>
> We sincerely appreciate your recognition of the high quality of our results, particularly their physical realism and our benchmark contribution.
>
> We believe there may be a few misunderstandings, and we are glad to address your concerns point by point.
>
> ---
> **[W1 & Q1]** ***[W1]:* Impractical Computational Cost: The "training-free" claim is misleading, as the method is a test-time optimization with extreme computational costs. Algorithm 1 and Table 5 indicate that it requires $k=10$ optimization iterations per denoising step $t$. For a 20-step process, this amounts to ~200 forward and backward passes. On a 12B parameter model like FLUX, this cost is computationally prohibitive and makes any comparison against single-forward-pass baselines impractical.** ***[Q1]:* Compute Cost Quantification: Could you provide a fair, wall-clock time comparison on identical hardware, e.g., how much time does SHINE (with $k=10$) take to generate one 1024x1024 image, versus the time required by baselines like AnyDoor or UniCombine?**
>
> Thank you for raising this important point. To address your concern clearly, we begin by (1) clarifying the misunderstanding regarding Table 5 and the term “training-free,” and then (2) provide the wall-clock comparison.
>
> 1. **Clarification Regarding Table 5 and the Meaning of “Training-Free”**
>
>     In brief, **our method performs optimization on only 3 denoising steps in the adapter setting and on only 2 denoising steps in the LoRA setting, rather than on all steps (see Table 5).** This means that the actual computational cost is substantially lower than what was assumed.
>
>     More specifically:
>
>     - *Scope of Optimization in Table 5:*
>     In Table 5, under *MSA loss*, “Step Range” indicates the denoising steps to which MSA optimization is applied. For the adapter setting, the MSA loss is applied only to the first 3 denoising steps (from $t = 14$ to $t = 12$). This corresponds to 30 forward passes. The most expensive backward passes are avoided by adopting the SDS strategy, as illustrated around Eqn. 2. (We further discuss the fundamental difference between our MSA loss and SDS in [W3]).
>
>     - *Interpretation of “Total Steps” in Table 5:*
>     In Table 5, under *Denoising Setup*, “Total steps” refers to the full diffusion/noising schedule, which defines a sequence of 20 values of $\sigma_t$ across timesteps $t$. However, our method does not begin denoising from the first timestep. As shown in Algorithm 1, it starts at $t_1 = 14$ (Table 5, Ours-Adapter), resulting in only 15 denoising steps in our process.
>
>     - *Lower Computational Cost for LoRA in Table 5:*
>     Since a LoRA is trained for a specific subject, it provides a more tailored prior than the generic adapter, allowing it to converge in fewer steps and with lower overall compute, as shown in Table 5.
>
>     - *Clarification of the term “training-free”:*
>     In our understanding, “training-free” usually means that no neural networks (e.g., the backbone diffusion model) are finetuned on the dataset for the task. This is different from being “optimization-free”, which means that the method does not perform any optimization over the latent.
>
> 2. **Wall-clock comparison**
>
>     Since several methods cannot directly handle high resolutions and most support 512×512 inputs, we report runtime and memory usage at 512×512. The wall-clock runtime on an H100 GPU is shown in the table below. In our implementation, we apply qint8 quantization to all FLUX-based methods to speed up inference and reduce GPU memory consumption. In addition, Ours-Adapter can also run on a 24-GB GPU by enabling CPU offloading.
>
>     |Method|Training-Free|Base|External Model|Time (s)|Peak Memory (MB)|
>     |-|:-:|:-:|:-:|:-:|:-:|
>     |MADD|No|SD|DINO|45.73|11708|
>     |ObjectStitch|No|SD|VIT|6.63|8268|
>     |DreamCom|No|SD|LoRA|9.87|3388|
>     |Anydoor|No|SD|DINO|8.61|18612|
>     |Unicombine|No|FLUX|LoRA|11.98|22711|
>     |PBE|No|SD|-|3.52|10842|
>     |TIGIC|Yes|SD|-|10.82|21640|
>     |TALE|Yes|SD|-|8.03|23524|
>     |TF-ICON|Yes|SD|-|24.55|20670|
>     |DreamEdit|Yes|SD|LoRA,VIT|99.83|19298|
>     |EEdit|Yes|FLUX|-|60.31|26546|
>     |Ours-Adapter|Yes|FLUX|Adapter|38.29|32552|
>     |Ours-LoRA|Yes|FLUX|LoRA|18.08|23519|
>
> We sincerely appreciate your question. Your confusion is likely due to our hyperparameter section being too concise, and we have expanded that part (Appendix J) to explain those hyperparameters, so that future readers can better understand it. We have also added the runtime comparison in Appendix N (or Table 8) of the revised version. Thank you again for your helpful feedback, which has allowed us to further improve the completeness and clarity of our work. Please feel free to let us know if you have further suggestions or thoughts. We’d be delighted to discuss them with you.

---

> ### Author Response · Authors · 2025-11-20
> **Response to Reviewer 6v7A (2/3)**
>
> **[W2 & Q3] *[W2]:* Brittle Pre-processing Pipeline: To avoid image inversion, the paper introduces a (Sec 3.1) pre-processing pipeline that depends on two external models: a VLM for captioning and an inpainting model for creating $x^{init}$. If the VLM caption is wrong (e.g., wrong color), the entire composition fails (as indicated in Sec 5-Limitation). It seems that this merely trades one failure point (inversion) for two new, uncontrolled failure points (VLM failure, inpainting failure)? *[Q3]:* Pre-processing Failure Rate: I am curious about the actual failure rate of your [VLM + Inpainting pre-processing pipeline], when running the 300-sample ComplexCompo benchmark.**
>
> Thank you for your thoughtful question and for highlighting this potential concern. We address your concerns below:
>
> 1. **Reliability of VLM color descriptions.** In practice, we find that the probability of the VLM (BLIP-3) producing an incorrect color description is extremely low. In Figure 8(b), the color mismatch is *constructed*: we intentionally modified the color term in the VLM-generated caption to a wrong color in order to illustrate the effect of such an error. Across both benchmarks, we did not observe any instance in which the VLM mis-identified the object color. Our motivation for exploring different inpainting prompts is to better understand how our algorithm behaves, so that we can provide clearer usage guidelines. When users supply their own inpainting prompts instead of using the VLM output, the key concern is to avoid inconsistencies in color descriptions.
>
> 2. **Robustness to inpainting imperfections.** As shown in Figure 8(a), inpainting can sometimes produce noticeable shape deviations or unnatural content, even under correct prompts. However, our algorithm can effectively refine such imperfect inpainting results during the optimization, so in practice the imperfections introduced by inpainting do not affect the final composition.
>
> 3. **Limitations of inversion-based approaches.** By contrast, the limitations of inversion are difficult to overcome. Inversion is only applicable to non-distilled models. For many modern distilled models (e.g., CFG-distilled or step-distilled variants), inversion and reconstruction accuracy can fall below 17 PSNR within 20 NFEs [1]. Moreover, inversion is slower than simply adding noise. These constraints motivate our choice to adopt the proposed pre-processing pipeline rather than relying on inversion.
>
> We hope these clarifications illustrate why the proposed pipeline is reliable in practice and why we prefer it over inversion-based alternatives. If you have further comments or suggestions, we would be happy to continue the conversation.
>
> [1] Deng, Y., He, X., Mei, C., Wang, P., & Tang, F. FireFlow: Fast Inversion of Rectified Flow for Image Semantic Editing. In Forty-second International Conference on Machine Learning (ICML 2025).

---

> ### Author Response · Authors · 2025-11-20
> **Response to Reviewer 6v7A (3/3)**
>
> **[W3 & Q2] *[W3]:* Limited Novelty and Model-Specific: MSA Loss: Re-naming a known technique (SDS) applied between an adapter and a base model does not constitute a significant novel contribution. DSG: This is an empirical, model-specific hack (via brute-force experimentation) for the FLUX architecture, it seems to lack a theoretical justification and guarantee of generalizability.** ***[Q2]* DSG Generalizability: You claim in Appendix E that DSG applies to SDXL, SD3.5, and PixArt. Was the "blur $Q_{img}$" heuristic applicable out-of-the-box to all these architectures, or did each one require a new round of brute-force experimentation to find a different hack (e.g., "blur $K_{txt}$")?**
>
> Thank you for your thoughtful question. We believe there may be a few misunderstandings, and we would be delighted to clarify them from two perspectives: (1) the difference between our MSA and SDS, and (2) why DSG is model-agnostic and generalizable.
>
> 1. **Difference Between MSA and SDS**
>    - Although MSA adopts the same computational simplification used in SDS (dropping the Jacobian term during backpropagation), **the two methods address fundamentally different problem settings and pursue distinct objectives.**
>
>    - SDS is primarily used to distill text-to-image priors into 3D representations (e.g., optimizing NeRF parameters). Its goal is to guide rendered images toward the diffusion model’s distribution by directly optimizing a diffusion loss comparing the model’s noise prediction to sampled noise. In this context, there is no requirement to preserve the geometric structure of a specific input image, so an anchor mechanism is unnecessary.
>
>    - In contrast, MSA is specifically designed for image composition. Our objective is to simultaneously (i) capture the subject’s identity and (ii) preserve the background so that the generated images exhibit natural interactions between the subject and the scene. To support this, we design an **anchor mechanism** that preserves the structural information of the background image.
>
>    - Furthermore, to the best of our knowledge, MSA is the first to introduce an open-domain customization adapter for guiding the optimization process, thereby ensuring robust preservation of the subject’s identity.
>
> 2. **On DSG and Its Generalizability**
>    - DSG, which applies blurring to $Q_{\text{img}}$, is mathematically equivalent to blurring the self-attention maps, as detailed in Line 308-309 and Appendix C. Because of this equivalence, DSG naturally generalizes to any architecture with a self-attention mechanism and does not require brute-force experimentation.
>
> We hope these explanations help address the concerns regarding novelty and generalizability. If you have further suggestions or would like us to elaborate on any part of the response, we would be glad to continue the discussion. Your feedback is genuinely helpful in strengthening our work.

---

### Author Response · Authors · 2025-12-01
**Summary of Our Rebuttal**

Dear Reviewers, ACs, SACs, and PCs,

Thank you for your time and dedication. We recognize that the workload is especially heavy this year. To help streamline the discussion, we provide a summary below. **We hope this would assist the ACs in quickly navigating the key issues addressed in our rebuttal and make the overall discussion easier to follow.**

**Strength:** As the reviewers noted, our approach produces high-quality, physically realistic compositions **(Yk7s, 6v7A)** and introduces a valuable benchmark **(6v7A)**. Our method and its main components are well-motivated and technically sound **(Yk7s, 5xsu)**, and the framework is both model-agnostic and practical **(5xsu)**.

---

**Below is a summary of our rebuttal points (Revisions in the paper have been highlighted in purple):**



**1. Response to Reviewer 6v7A**

- [W1 & Q1] We clarified the reviewer’s misunderstanding about our algorithm, explaining that optimization is applied to only 2–3 denoising steps rather than every step. We also provided runtime and memory comparisons.

- [W2 & Q3] We confirmed the robustness of our preprocessing pipeline (VLM + inpainting).

- [W3 & Q2] We explained the difference between our MSA and SDS, and further clarified the reviewer’s misconception about our DSG, demonstrating that DSG is generalizable, as detailed in Appendix C.

Due to time constraints, Reviewer 6v7A may not have had the chance to respond or indicate any further concerns.

---
**2. Response to Reviewer 5xsu**

- [W1] We adopted the terminology provided by Reviewer 5xsu.

- [W2] We clarified our rationale for using one-step forward diffusion instead of inversion, and further discussed how our method differs from EEdit.

- [W3] We added new experiments comparing the negative prompt and our DSG on SD3.5.

- [W4] We included details about the inpainting model and the VLM used in our pipeline.

- [W5] We expanded Algorithm 1 with additional explanations.

- [W6] We added a baseline to Tables 1 and 3.

- [Q1] We provided a description of how 2D Gaussian blur is implemented in our framework.

- [Q2] We provided runtime and memory comparisons.

- [Follow-up] We improved the bibliography and presentation of Sections C and K.

As a result, Reviewer 5xsu expressed willingness to increase the score **from 8 to 10**.

---

**3. Response to Reviewer Yk7s**

- [W1] We provided insights into why background retention cannot consistently achieve SOTA across all cases, and outlined two possible directions for future improvement.

- [W2] We added further evaluation using more recent reward models.

- [Follow-up] We corrected a typo identified by Reviewer Yk7s.

As a result, Reviewer Yk7s confirmed that **the concerns have been adequately addressed**.

---

With sincere appreciation,

The Authors

---

### Meta-Review · Area_Chair_bMG5 · 2026-01-06

**Summary:**

Reviewer 6v7A: Main concerns are the very high computational cost of the “training-free” test-time optimization, brittleness of the VLM + inpainting preprocessing pipeline, and limited methodological novelty (MSA and DSG viewed as incremental/model-specific).

Reviewer 5xsu: While strongly positive overall, requests clarifications and deeper analysis on terminology, the one-step diffusion/inversion assumption, the behavior and generality of DSG vs. negative prompting, details of the external models used, and clearer algorithmic notation and benchmarking.

Reviewer Yk7s: Notes room for improvement in Background Retention and recommends broader image-quality evaluation using additional preference models.

These concerns have been largely addressed by the rebuttal and are balanced by strong qualitative performance, solid technical insight, and meaningful practical contributions.

**Reviewer Concerns:**

Most of the reviewers’ concerns have been addressed.

Reviewer Yk7s noted that, as shown in Table 1, the method performs well in Subject Identity Consistency, but there remains room for improvement in Background Retention, and asked for further insights and possible avenues for enhancement. The authors explained why background retention may not consistently achieve state-of-the-art performance across all cases and outlined two potential directions for future improvement.

Although this concern is not fully resolved, Reviewer Yk7s is satisfied with the response and is comfortable leaving it as future work.

**Reviewer Scores:**

Reviewer 6v7A: 6
Reviewer 5xsu: 8
Reviewer Yk7s: 8

The authors have carefully addressed the reviewers’ concerns. Both Reviewer 5xsu and Reviewer Yk7s expressed satisfaction with the rebuttal; in particular, Reviewer 5xsu raised the score from 8 to 10. I also expect that Reviewer 6v7A will at least maintain the original score.

---

### Decision · Program_Chairs · 2026-01-26

Accept (Poster)